# Collaborative Discrete-Continuous Black-Box Prompt Learning for Language Models

**Hualin Zhang**[1]∗  **Haozhen Zhang**[2]∗  **Zhekai Liu**[3]∗  **Bin Gu**[2]†  **Yi Chang**[2,4,5]†

[1]Department of Machine Learning, Mohamed bin Zayed University of Artificial Intelligence
[2]School of Artificial Intelligence, Jilin University [3]School of Mathematics, Jilin University
[4]International Center of Future Science, Jilin University
[5]Engineering Research Center of Knowledge-Driven Human-Machine Intelligence, MOE
`zhanghualin98@gmail.com haozhen23@mails.jlu.edu.cn Liu_Zhekai@outlook.com`
`jsgubin@gmail.com yichang@jlu.edu.cn`

## ABSTRACT

Large Scale Pre-Trained Language Models (PTMs) have demonstrated unprecedented capabilities across diverse natural language processing tasks. Adapting such models to downstream tasks is computationally intensive and time-consuming, particularly in black-box scenarios common in Language-Model-as-a-Service (LMaaS) environments, where model parameters and gradients are inaccessible. Recently, black-box prompt learning using zeroth-order gradients has emerged as a promising approach to address these challenges by optimizing learnable continuous prompts in embedding spaces, starting with *randomly initialized discrete text prompts*. However, its reliance on randomly initialized discrete prompts limits adaptability to diverse downstream tasks or models. To address this limitation, this paper introduces ZO-PoG, a novel framework that optimizes prompts through a collaborative approach, combining Policy Gradient optimization for initial discrete text prompts and Zeroth-Order optimization for continuous prompts in embedding space. By optimizing collaboratively between discrete and continuous prompts, ZO-PoG maximizes adaptability to downstream tasks, achieving superior results without direct access to the model's internal structures. Importantly, we establish the sub-linear convergence of ZO-PoG under mild assumptions. The experiments on different datasets demonstrate significant improvements in various tasks compared to the baselines. Our code is available at: https://github.com/zhanghualin0/ZO-PoG.

## 1 INTRODUCTION

Large-scale Pre-trained Language Models (PTMs) have demonstrated exceptional capabilities across a broad spectrum of natural language processing tasks (Devlin, 2018; Raffel et al., 2020; Brown, 2020; Zhang et al., 2021; Zeng et al., 2021; Sun et al., 2021; Fedus et al., 2022), particularly since the advent of models like GPT-3 by (Brown, 2020). Their popularity stems from their ability to generalize to various downstream tasks with minimal supervision, often through few-shot or zero-shot learning.

However, commercial models often limit users to interacting with PTMs solely via APIs, offering no access to the underlying parameters or gradients so we cannot optimize the model by the traditional method. In reality, adapting PTMs to specific tasks remains computationally intensive, especially in black-box scenarios common in Language-Model-as-a-Service (LMaaS) environments (Zhao et al., 2022; Sun et al., 2022b), where direct access to model parameters and gradients is restricted. Moreover, the growing size and complexity pose significant challenges for practical adaptation and deployment. Fine-tuning PTMs requires extensive computational resources and access to model parameters, which is often impractical for many users due to cost and privacy concerns.

---

∗Equal contribution.
†Corresponding authors.

To overcome these challenges, black-box prompt learning has emerged as a promising solution. This approach seeks to modify and optimize input prompts to improve task performance while avoiding the need for full model access or fine-tuning. Prompt learning focuses on crafting or optimizing input prompts to guide the PTM in producing task-relevant outputs. A successful prompt can transform the model's behavior on various downstream tasks, improving performance without modifying the model. In scenarios where only function evaluations are available, derivative-free optimization (DFO) algorithms have been widely used. In a black-box setting, this optimization needs to be derivative-free, as gradients are not available.

Building on these insights, several approaches for black-box prompt learning have emerged, such as Black-Box Tuning (BBT) (Sun et al., 2022a), which optimizes continuous prompts through derivative-free methods in embedding space. Since the dimensionality of the original embedding space could be tens of thousands while DFO methods suffer from high variance in high-dimensional space. BBT proposed to solve this problem by optimizing the continuous prompts in a low-dimensional subspace based on the fact that common PTMs already have a low intrinsic dimensionality (Li et al., 2018; Aghajanyan et al., 2020). This leads to an effective way of utilizing DFO algorithms for black-box prompt learning. Despite these advances, these methods encounter notable limitations.

Continuous prompt tuning often starts with randomly initialized discrete prompts, which can result in suboptimal performance and limit adaptability across different tasks and models. The discrete-to-continuous transfer lacks guidance from task-specific signals, reducing the efficiency and adaptability of the learned prompts. Besides optimizing continuous prompt in embedding space, recent research has also proposed effective methods for optimizing discrete prompts directly (Deng et al., 2022; Diao et al., 2022). These approaches have demonstrated promising results, offering pathways to enhance discrete prompt optimization. However, while these methods offer a structured way to optimize discrete prompts, they often struggle to efficiently translate task-specific information into the continuous embeddings used by language models or to effectively guide model behavior in black-box settings where internal parameters are inaccessible.

This brings us to a critical question: can we harness the strengths of both discrete and continuous prompt optimization to create a more adaptable and robust prompting framework? In order to tackle the above problem, this paper presents zeroth-order and policy gradient method (ZO-PoG) - a collaborative framework that alternates between optimizing discrete prompts and continuous prompts in low-dimensional intrinsic spaces. For discrete prompt optimization, we first reparamterize the categorical distribution of the PTM vocabulary with Gumbel-Softmax trick (Maddison et al., 2016; Jang et al., 2016) and use the policy gradient method (Diao et al., 2022) for gradient approximation in the black-box setting. This allows us to optimize in parameter space rather than token probabilities, mitigating bias. For continuous prompt optimization, we use zeroth-order gradient to optimize in the lower-dimensional subspace by using a random projection. By doing so, ZO-PoG bridges the gap between discrete and continuous prompt learning, ensuring that prompts are tailored to maximize performance across diverse downstream tasks in black-box environments.

The main contributions of this paper are summarized as follows:

- We introduce ZO-PoG, a prompt learning framework that alternates between optimizing discrete and continuous prompts. This approach is the first to jointly optimize both discrete text prompts and the continuous embeddings for black-box PTMs, enhancing the adaptability and performance of the prompts across different downstream tasks.

- We formally establish the convergence of our framework, showing that ZO-PoG achieves a sub-linear convergence rate under mild assumptions. Our analysis highlights how variance and biases are controlled by hyperparameters and mini-batch sizes, ensuring that our method remains effective for tuning PTMs without accessing model parameters.

- Through extensive experiments on various datasets, we demonstrate that ZO-PoG significantly improves the performance of PTMs. Our findings also underscore the importance of prompt initialization, showing that an optimized starting prompt is crucial for achieving better performance, thereby enhancing the overall effectiveness of our collaborative optimization strategy.

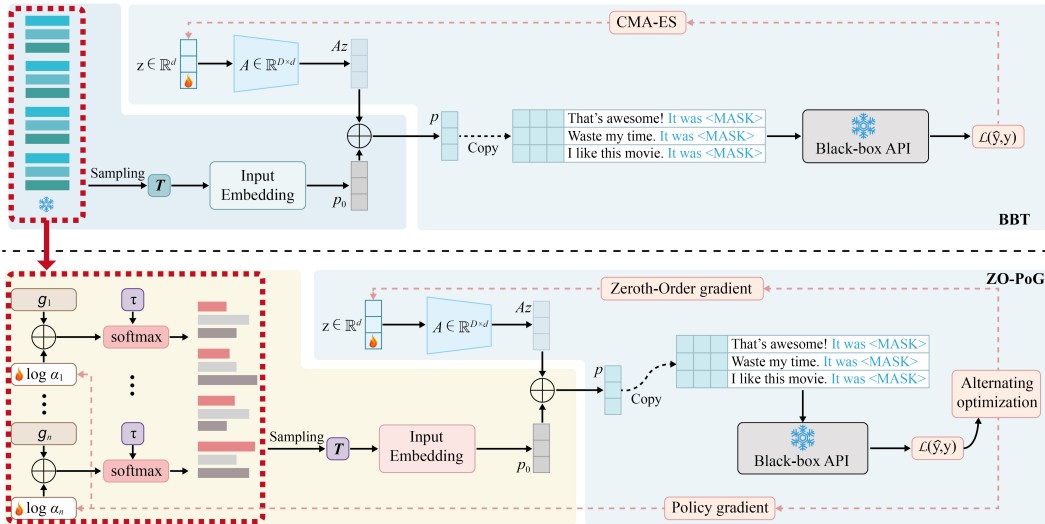

Figure 1: **Overview of BBT (Up) and the proposed ZO-PoG framework (Down).** Alternating optimization process between discrete and continuous prompt tuning in the ZO-PoG framework. The discrete prompts are first optimized through policy gradient in the parameter space to enhance interpretability and task-specific alignment. Subsequently, the continuous prompt representations are refined using zeroth-order gradient optimization in a low-dimensional intrinsic space.

## 2    RELATED WORK

**Language Models as a Service.** Language Models as a Service (LMaaS), which was early introduced by Zhao et al. (2022); Sun et al. (2022b), describes a diagram where models are accessible through APIs without direct access to their internal parameters. This paradigm has grown in prominence with systems like OpenAI's ChatGPT, Google's PaLM. Recently, several lines of research in the scope of LMaaS has focused on adapting pre-trained models to various downstream tasks. One approach is through text prompting, where manually or automatically designed prompts help tailor model outputs (Brown, 2020; Kojima et al., 2022). In-context learning further enhances this by providing labeled examples within prompts to quickly adapt the model (Chen et al., 2021; Xie et al., 2021). The capabilities of PTMs also evaluated across multiple benchmarks (Liang et al., 2022; Chang et al., 2024; Bubeck et al., 2023).

**Black-box Prompt Learning.** Black-box prompt learning methods aim to modify input prompts to guide the PTM toward producing task-relevant outputs. A key challenge lies in optimizing these prompts without access to gradients or internal parameters, leading to reliance on derivative-free optimization (DFO) algorithms. One line of this kind of works focused on optimizing the soft prompts in continuous embedding space. One notable work in this area is Black-Box Tuning (BBT) (Sun et al., 2022b), which optimizes soft prompt with Covariance Matrix Adaptation Evolution Strategy (CMA-ES). BBT addressed the challenge of high dimensionality in the embedding space by focusing on a lower-dimensional subspace, leveraging the low intrinsic dimensionality of PTMs. BBTv2 (Sun et al., 2022a), an improved version of BBT, optimizes prompts across all layers of a pre-trained language model using a gradient-free, divide-and-conquer algorithm, achieving few-shot learning performance comparable to full model tuning with significantly fewer tunable parameters. Another line of work focused on optimizing discrete prompt tokens in black-box scenarios, which is more suitable for black-box large language models with inference APIs like ChatGPT, GPT-4, Claude, etc. These work can mainly be categorized by reinforcement learning (Deng et al., 2022; Zhang et al., 2023), ensemble (Hou et al., 2023), iterative search (Prasad et al., 2022), policy gradient optimization (Diao et al., 2022), Bayesian optimization for instruction optimization(Chen et al., 2024; Hu et al., 2024), multi-armed bandit (Pryzant et al., 2023; Lin et al., 2023; Shi et al., 2024).

## 3 METHODOLOGY

In this section, we start by introducing the problem formulation in Section 3.1, where we define the task of optimizing prompts for a pre-trained language model (PTM) in a black-box setting. Specifically, we focus on optimizing input prompts to improve the model's performance on downstream tasks. Then, in Section 3.2, we present the discrete prompt optimization approach using policy gradient estimations to find optimal prompts over the vocabulary space. Next, we proceed to Section 3.3, where we describe continuous prompt learning through zeroth-order gradient (ZO), which allows for fine-grained adjustments in the prompt embeddings without requiring access to gradients. Finally, we present our alternating optimization strategy in Section 3.4, where we collaboratively optimize both discrete and continuous prompts, enabling effective performance on downstream tasks.

**Notations.** Throughout this paper, we use bold uppercase letters (e.g., $\mathbf{A}$) to denote matrices and bold lowercase letters (e.g., $\mathbf{z}$) to denote vectors. We use $T = [t_1, t_2, \ldots, t_n]$ to represent the prompt tokens of length $n$ and each token $t_i$ is chosen from the PTM vocabulary $\mathcal{V}$. We use $p = [p_1, \ldots, p_n] \in \mathbb{R}^{|\mathcal{V}| \times n}$ to denote the probability distribution of the prompt tokens over the vocabulary. Each $p_i$ should satisfy the categorical distribution, i.e., $\forall i = 1, \ldots, n, p_i \in \mathcal{C} := \{p : \|p\|_1 = 1, 0 \preceq p \preceq 1\}$. We use $\mathbf{e}(\cdot) : \mathcal{V}^n \to \mathbb{R}^D$ to denote the map from prompt tokens of length $n$ to the prompt embedding of dimension $D$. We use $\|\cdot\|$ to denote the Euclidean norm of a vector.

### 3.1 PROBLEM FORMULATION

Given a pre-trained language model (PTM) in a black-box scenario, we aim to optimize the input prompts to enhance the model's performance on specific downstream tasks. Specifically, for a batch of primitive texts $X$ and labels $Y$, the general goal of prompt learning is to find the optimal prompt $\mathbf{p}^* = \arg\min_{\mathbf{p} \in \mathbb{R}^D} \mathcal{L}(f(\mathbf{p}; X), Y)$, where $D$ is the dimensionality of the prompt embedding space and $f$ represents a black-box PTM where the inputs are combined with learnable prompts and the primitive texts $X$. Directly optimizing the prompt in the embedding space is impractical due to the high dimensionality, which can reach tens of thousands. Sun et al. (2022b) proposed to optimize over the subspace of the original embedding space with much lower intrinsic dimensionality. Specifically, the objective is

$$\mathbf{z}^* = \arg\min_{\mathbf{z}} \mathcal{L}(f(\mathbf{A}\mathbf{z} + \mathbf{p}_0; X), Y),$$

where $\mathbf{z} \in \mathbb{R}^d$, $\mathbf{A} \in \mathbb{R}^{D \times d}$ is the projection matrix (with $d \ll D$), $\mathcal{D} = \{X, Y\}$ is the total dataset, and $\mathbf{p}_0 \in \mathbb{R}^D$ is the initial prompt, which are embedded tokens chosen randomly from the PTM vocabulary.

However, random initialization may lead to a suboptimal prompt. A more effective approach is to optimize the initial prompt $\mathbf{p}_0$. Directly optimizing the initial prompt in the embedding space presents the same challenge of high dimensionality. To address this, we propose to optimize the initial prompt $\mathbf{p}_0$ in the discrete probability space of the PTM vocabulary, rather than the continuous embedding space. Specifically, instead of selecting prompt tokens uniformly at random from the PTM vocabulary, we aim to learn a distribution over the vocabulary, allowing for more targeted and meaningful token selection. The details of the optimization method of the probability distribution will be discussed in Section 3.2.

Based on the above considerations, we will optimize the initial discrete prompt $\mathbf{p}_0$ by learning a probability distribution over the vocabulary rather than selecting prompt tokens randomly. This allows us to improve the selection of tokens and guide the model more effectively in black-box settings. Following this, we will alternate between optimizing the discrete prompt $\mathbf{p}_0$ and optimizing the continuous prompt $\mathbf{z}$ in the lower-dimensional intrinsic space, ensuring an efficient and scalable solution to prompt learning in black-box scenarios. The main objective of this work becomes:

$$\min_{P(T), \mathbf{z}} \mathbb{E}_{T \sim P(T)} \mathcal{L}\left(f(\mathbf{A}\mathbf{z} + \mathbf{p}_0; X), Y\right) = \mathbb{E}_{T \sim P(T)} \mathcal{L}(f(\mathbf{A}\mathbf{z} + \mathbf{e}(T); X), Y). \tag{1}$$

In the above objective, the prompt $\mathbf{p}$ can be decomposed into three components: an initial discrete prompt embedding $\mathbf{p}_0 \in \mathbb{R}^D$ obtained by sampling $n$ tokens from the PTM vocabulary with a trainable distribution, a continuous intrinsic low dimensional vector $\mathbf{z} \in \mathbb{R}^d$, and a random matrix $\mathbf{A} \in \mathbb{R}^{D \times d}$ that projects $\mathbf{z}$ back to the original high dimensional prompt embedding space.

### 3.2 Black-Box Discrete Prompt Learning with Policy gradient

To optimize discrete text prompts in black-box scenarios, we adopt a policy gradient approach. Specifically, we aim to optimize the distribution of discrete prompt tokens over a given vocabulary, where each token is represented by a probability distribution. Let $p = [p_1, \ldots, p_n] \in \mathbb{R}^{|\mathcal{V}| \times n}$ denote the probability distribution of $n$ discrete prompt tokens over the vocabulary $\mathcal{V}$ of size $|\mathcal{V}|$, where $p_i \in \mathcal{C} := \{p : \|p\|_1 = 1, 0 \preceq p \preceq 1\}$.

**Gumbel-Softmax Trick for Smoothing Optimization.** Instead of directly manipulating the probability distribution, we reparameterize the categorical distribution using the Gumbel-Softmax trick (Jang et al., 2016; Maddison et al., 2016). This allows us to transform the optimization process into one over continuous parameters, reducing bias and ensuring smooth optimization. To be specific,

$$p_{i,j} = \text{softmax}\left(\frac{\log(\boldsymbol{\alpha}_{i,j}) + g_{i,j}}{\tau}\right) = \frac{\exp\left(\frac{\log(\boldsymbol{\alpha}_{i,j}) + g_{i,j}}{\tau}\right)}{\sum_{l=1}^{|\mathcal{V}|} \exp\left(\frac{\log(\boldsymbol{\alpha}_{i,l}) + g_{i,l}}{\tau}\right)}, \tag{2}$$

where $\boldsymbol{\alpha}_{i,j}$ denotes the $j$-th component of a column vector $\boldsymbol{\alpha}_i$, $g_{i,j}$ is the Gumbel random variable, which is sampled from the Gumbel$(0, 1)$ distribution and $\tau$ is the temperature parameter controlling the smoothness of the approximation. After reparameterization, the optimization objective becomes:

$$\min_{\boldsymbol{\alpha}, \mathbf{z}} \mathbb{E}_{T \sim P(T|\boldsymbol{\alpha})} \mathcal{L}(f(\mathbf{A}\mathbf{z} + \mathbf{e}(T); X), Y). \tag{3}$$

For abbreviation, we denote $\mathcal{L}(f(\mathbf{A}\mathbf{z} + \mathbf{e}(T); X), Y)$ as $\mathcal{L}(T)$ for fixed $\mathbf{z}$ in the rest of this subsection.

Since accessing the model's internal parameters is often restricted, the problem involves optimizing the discrete probability distribution with just forward passes through the black-box models. To achieve this, we will adopt the policy gradient estimator (PGE) to optimize the above objective. According to the policy gradient theorem (Sutton et al., 1999), we can estimate the parameter's gradient with respect to $\boldsymbol{\alpha}_i$ by:

$$\begin{aligned}
\nabla_{\boldsymbol{\alpha}_i} \mathbb{E}_T[\mathcal{L}(T)] &= \sum_{T \in \mathcal{V}^n} \mathcal{L}(T) \nabla_{\boldsymbol{\alpha}_i} P(T|\boldsymbol{\alpha}) = \sum_{T \in \mathcal{V}^n} \mathcal{L}(T) \frac{P(T|\boldsymbol{\alpha})}{P(T|\boldsymbol{\alpha})} \nabla_{\boldsymbol{\alpha}_i} P(T|\boldsymbol{\alpha}) \\
&= \sum_{T \in \mathcal{V}^n} P(T|\boldsymbol{\alpha}) \mathcal{L}(T) \nabla_{\boldsymbol{\alpha}_i} \log P(T|\boldsymbol{\alpha}) = \mathbb{E}_{P(T|\boldsymbol{\alpha})}[\mathcal{L}(T) \nabla_{\boldsymbol{\alpha}_i} \log \Pi_{j=1}^n P(t_j|\boldsymbol{\alpha}_j)] \\
&= \mathbb{E}_{P(T|\boldsymbol{\alpha})}[\mathcal{L}(T) \nabla_{\boldsymbol{\alpha}_i} \log P(t_i|\boldsymbol{\alpha}_i)].
\end{aligned}$$

**Variance Reduction in Policy Gradient Estimation.** One of the primary challenges in applying policy gradient methods is the high variance in gradient estimates, which can lead to unstable updates and slow convergence. To address this, we implement a variance-reduced policy gradient estimator (VR-PGE) (Williams, 1992; Dong et al., 2020; Zhou et al., 2021; Diao et al., 2022), which reduces variance by subtracting a baseline from the reward function during updates. The estimated gradient is calculated by:

$$g_{\boldsymbol{\alpha}_i}^{vr} = \frac{1}{I-1} \sum_{k=1}^{I} \left(\mathcal{L}(T^{(k)}) - \frac{1}{I} \sum_{j=1}^{I} \mathcal{L}(T^{(j)})\right) \nabla_{\boldsymbol{\alpha}_i} \log P(t_i|\boldsymbol{\alpha}_i), \tag{4}$$

where $T^{(k)}, k = 1, \cdots, I$ are sampled independently from $P(T|\boldsymbol{\alpha})$ with parameter $\boldsymbol{\alpha} \in \mathbb{R}^{|\mathcal{V}| \times n}$.

### 3.3 Black-Box Continuous Prompt Learning with Zeroth-Order gradient

In addition to optimizing discrete prompts described in Section 3.2, continuous prompts provide a more fine-grained mechanism to guide pre-trained models (PTMs) in black-box settings. Given that the internal gradients of black-box models are inaccessible, zeroth-order gradient (ZO) (Nesterov & Spokoiny, 2017) is an effective approach for optimizing continuous prompts without needing gradient information.

For abbreviation, we denote $\mathcal{L}(f(\mathbf{A}\mathbf{z} + \mathbf{e}(T); X), Y)$ as $\mathcal{L}(\mathbf{z})$ for fixed $\mathbf{e}(T)$ in the rest of this subsection. Then, we aim to optimize a low-dimensional continuous vector $\mathbf{z} \in \mathbb{R}^d$ in the intrinsic

subspace of the prompt embedding space. The ZO gradient estimator approximates the gradient of the loss function using a symmetric difference:

$$g_{\mathbf{z}} = \frac{\mathcal{L}(\mathbf{z} + \mu\mathbf{u}) - \mathcal{L}(\mathbf{z} - \mu\mathbf{u})}{2\mu}\mathbf{u}, \qquad (5)$$

where $\mu > 0$ is the smoothing parameter and $\mathbf{u} \in \mathbb{R}^d$ is distributed in $\mathcal{N}(0, \mathbf{I}_d)$.

### 3.4 ALTERNATING OPTIMIZATION BETWEEN DISCRETE AND CONTINUOUS PROMPTS

In this section, we describe our proposed optimization algorithm, which alternates between optimizing discrete and continuous prompts. This method takes advantage of the unique properties of both prompt types to effectively enhance performance in black-box models. The algorithm leverages the strengths of policy gradient for discrete prompt optimization and zeroth-order gradient estimation for continuous prompts, achieving efficient prompt learning in black-box settings.

In each iteration, the optimization process alternates between two key steps: 1) Discrete Prompt Optimization: This involves learning a distribution over discrete prompt tokens from the model's vocabulary. Each token in the prompt is sampled from a Gumbel-Softmax distribution, and policy gradient methods are used to update the distribution parameters $\boldsymbol{\alpha} \in \mathbb{R}^{|\mathcal{V}| \times n}$. 2) Continuous Prompt Optimization: After discrete prompts are optimized, the continuous prompt, represented as a low-dimensional vector $\mathbf{z} \in \mathbb{R}^d$ in the prompt embedding space, is optimized using zeroth-order gradient estimation. This is crucial in the black-box setting where internal gradients are unavailable.

By alternating between discrete prompt optimization and continuous prompt optimization ensures that both the categorical structure of tokens and the fine-grained continuous embeddings are optimized in tandem. This complementary approach maximizes the model's performance on downstream tasks by balancing discrete token selection with smooth embedding adjustments.

---

**Algorithm 1** Black-Box Prompt Learning via Zeroth-Order and Policy Gradient Method

---

**Input:** Parameters of categorical distributions $\boldsymbol{\alpha}_1, \cdots, \boldsymbol{\alpha}_n$, temperature parameter $\tau$, prediction model $f$, PTMs embedding function $\mathbf{e}(\cdot)$, downstream dataset $\mathcal{D}$, mini-batch size $B$, learning rates $\eta_{\boldsymbol{\alpha}}, \eta_{\mathbf{z}}$ and sample times $I_1, I_2$.
1: Initialize $\boldsymbol{\alpha}_1^0, \cdots, \boldsymbol{\alpha}_n^0, \mathbf{z}^0$
2: **for** $t$ in 0 to $T - 1$ **do**
3:      Draw a mini-batch $\mathcal{S}_t = \{X_t, Y_t\} = \{x_i, y_i\}_{i=1}^B$ from $\mathcal{D}$
4:      **for** $k$ in 1 to $I_1$ **do**
5:          Sample $j_{1,t}^{(k)} \sim \text{GS}(\boldsymbol{\alpha}_1^t, \tau), \cdots, j_{n,t}^{(k)} \sim \text{GS}(\boldsymbol{\alpha}_n^t, \tau)$
6:          $\mathbf{p}_{0,t}^{(k)} = \mathbf{e}(t_1^{(k)}, \cdots, t_n^{(k)}) = \mathbf{e}(\mathcal{V}[j_1^{(k)}], \cdots, \mathcal{V}[j_n^{(k)}])$
7:      $\mathcal{L}_{\text{avg}} = \frac{1}{I_1}\sum_{k=1}^{I_1} \mathcal{L}(f(\mathbf{A}\mathbf{z}^t + \mathbf{p}_{0,t}^{(k)}; X_t), Y_t)$
8:      **for** $i$ in 1 to $n$ **do**
9:          $g_{\boldsymbol{\alpha}_i^t}^{vr} = \frac{1}{I_1 - 1}\sum_{k=1}^{I_1}(\mathcal{L}(f(\mathbf{A}\mathbf{z}^t + \mathbf{p}_{0,t}^{(k)}; X_t), Y_t) - \mathcal{L}_{\text{avg}})\nabla_{\boldsymbol{\alpha}_i} \log P(t_i^{(k)} | \boldsymbol{\alpha}_i^t)$
10:          $\boldsymbol{\alpha}_i^{t+1} \leftarrow \boldsymbol{\alpha}_i^t - \eta_{\boldsymbol{\alpha}} g_{\boldsymbol{\alpha}_i^t}^{vr}$               ▷ Update discrete vocabulary distribution
11:      **for** $l$ in 1 to $I_2$ **do**
12:          Sample $j_{1,t}^{(l)} \sim \text{GS}(\boldsymbol{\alpha}_1^{t+1}, \tau), \cdots, j_{n,t}^{(l)} \sim \text{GS}(\boldsymbol{\alpha}_n^{t+1}, \tau)$
13:          $\mathbf{p}_{0,t}^{(l)} = \mathbf{e}(t_{1,t}^{(l)}, \cdots, t_{n,t}^{(l)}) = \mathbf{e}(\mathcal{V}[j_{1,t}^{(l)}], \cdots, \mathcal{V}[j_{n,t}^{(l)}])$
14:          Sample $\mathbf{u}_t^{(l)} \sim \mathcal{N}(0, \mathbf{I}_d)$
15:          $g_{\mathbf{z}^t}^{(l)} = \frac{\mathcal{L}(f(\mathbf{A}(\mathbf{z}^t + \mu\mathbf{u}_t^{(l)}) + \mathbf{p}_{0,t}^{(l)}; X_t), Y_t) - \mathcal{L}(f(\mathbf{A}(\mathbf{z}^t - \mu\mathbf{u}_t^{(l)}) + \mathbf{p}_{0,t}^{(l)}; X_t), Y_t)}{2\mu}\mathbf{u}_t^{(l)}$
16:      $g_{\mathbf{z}^t} = \frac{1}{I_2}\sum_{l=1}^{I_2} g_{\mathbf{z}^t}^{(l)}$
17:      $\mathbf{z}^{t+1} = \mathbf{z}^t - \eta_{\mathbf{z}} g_{\mathbf{z}^t}$               ▷ Update continuous prompt
**Return:** $\boldsymbol{\alpha}_1^T, \cdots, \boldsymbol{\alpha}_n^T, \mathbf{z}^T$

---

# 4 CONVERGENCE ANALYSIS

In this section, we provide a detailed analysis of the convergence behavior of our proposed framework. Our goal is to demonstrate the convergence properties of the alternating optimization process that integrates the policy gradient method for discrete prompt learning and the zeroth-order method for continuous prompt optimization.

We begin by establishing several basic assumptions about the smoothness and bounded variance of the loss function. Under these assumptions, we derive the convergence rate of our method and analyze the impact of both the policy gradient and zeroth-order gradient on the overall convergence. Specifically, we make the following assumptions:

**Assumption 1** (Smoothness). *For convenience, we will use $\mathcal{L}(\boldsymbol{\alpha}, \mathbf{z})$ to represent the loss function $\mathcal{L}(f(\mathbf{A}\mathbf{z} + \mathbf{p}_0; X), Y)$, where $\mathcal{D} = \{X, Y\}$ is the whole dataset, $f$ is the prediction function, and $\mathbf{p}_0$ is sampled from the token distribution which is generated by $\boldsymbol{\alpha}$. Suppose that $\mathcal{L}(\cdot)$ is block-wise smooth with gradient Lipschitz constant $L_{\boldsymbol{\alpha}}$, that is for $\forall \boldsymbol{\alpha}, \boldsymbol{\alpha}' \in \mathbb{R}^{|\mathcal{V}| \times n}, \mathbf{z} \in \mathbb{R}^d$ :*

$$\|\nabla_{\boldsymbol{\alpha}}\mathcal{L}(\boldsymbol{\alpha}, \mathbf{z}) - \nabla_{\boldsymbol{\alpha}}\mathcal{L}(\boldsymbol{\alpha}', \mathbf{z})\| \leq L_{\boldsymbol{\alpha}}\|\boldsymbol{\alpha} - \boldsymbol{\alpha}'\|. \tag{6}$$

*and block-wise smooth with gradient Lipschitz constant $L_{\mathbf{z}}$, that is for $\forall \mathbf{z}, \mathbf{z}' \in \mathbb{R}^d, \boldsymbol{\alpha} \in \mathbb{R}^{|\mathcal{V}| \times n}$ :*

$$\|\nabla_{\mathbf{z}}\mathcal{L}(\boldsymbol{\alpha}, \mathbf{z}) - \nabla_{\mathbf{z}}\mathcal{L}(\boldsymbol{\alpha}, \mathbf{z}')\| \leq L_{\mathbf{z}}\|\mathbf{z} - \mathbf{z}'\|. \tag{7}$$

**Assumption 2** (Bounded Variance). *Assume that the stochastic gradient has bounded variance, i.e.,*

$$\mathbb{E}_{(x_i, y_i) \in \mathcal{D}}\|\nabla_{\boldsymbol{\alpha}}\mathcal{L}_i(\boldsymbol{\alpha}, \mathbf{z}) - \nabla_{\boldsymbol{\alpha}}\mathcal{L}(\boldsymbol{\alpha}, \mathbf{z})\|^2 \leq \sigma_{\boldsymbol{\alpha}}^2, \quad \forall \mathbf{z} \in \mathbb{R}^d; \tag{8}$$

$$\mathbb{E}_{(x_i, y_i) \in \mathcal{D}}\|\nabla_{\mathbf{z}}\mathcal{L}_i(\boldsymbol{\alpha}, \mathbf{z}) - \nabla_{\mathbf{z}}\mathcal{L}(\boldsymbol{\alpha}, \mathbf{z})\|^2 \leq \sigma_{\mathbf{z}}^2, \quad \forall \boldsymbol{\alpha} \in \mathbb{R}^{|\mathcal{V}| \times n}. \tag{9}$$

**Assumption 3.** *There exists a positive constant $C < \infty$ such that $|\mathcal{L}_{\mathcal{S}}(\boldsymbol{\alpha}, \mathbf{z})| < C, \forall \boldsymbol{\alpha} \in \mathbb{R}^{|\mathcal{V}| \times n}, \mathbf{z} \in \mathbb{R}^d$.*

**Assumption 4.** *In the policy gradient method, we optimize $\boldsymbol{\alpha}_i$, where $i$ is from $1$ to $n$. The elements in parameter vector $\boldsymbol{\alpha}_i$ can be written by $\boldsymbol{\alpha}_{i,j}$ has a lower bound denote: $0 \leq \varepsilon \leq \boldsymbol{\alpha}_{i,j}$.*

**Remark 1.** *In our experiments, after each iteration, we truncate elements $\boldsymbol{\alpha}_{i,j}$ to $1 \times 10^{-11}$ if it falls below this threshold. This precaution prevents the gradient $\boldsymbol{\alpha}_i$ from becoming excessively large or meaningless during optimization, which can occur due to the properties of the logarithm.*

**Proposition 1.** *Under Assumption 2, 3 and 4, the stochastic policy gradient $g_{\boldsymbol{\alpha}_i}^{vr}$ has a bounded variance*

$$\mathbb{E}_{\mathcal{S}}\mathbb{E}_{T \sim P(T)}\|g_{\boldsymbol{\alpha}_i}^{vr} - \nabla_{\boldsymbol{\alpha}_i}\mathcal{L}(\boldsymbol{\alpha}, \mathbf{z})\|^2$$

$$\leq 2\underbrace{\mathbb{E}_{T \sim P(T)}\|g_{\boldsymbol{\alpha}_i}^{vr} - \nabla_{\boldsymbol{\alpha}_i}\mathcal{L}_{\mathcal{S}}(\boldsymbol{\alpha}, \mathbf{z})\|^2}_{\text{Where } \mathcal{S} \text{ is fixed}} + 2\underbrace{\mathbb{E}_{\mathcal{S}}\|\nabla_{\boldsymbol{\alpha}_i}\mathcal{L}_{\mathcal{S}}(\boldsymbol{\alpha}, \mathbf{z}) - \nabla_{\boldsymbol{\alpha}_i}\mathcal{L}(\boldsymbol{\alpha}, \mathbf{z})\|^2}_{\text{Variance from mini-batch } \mathcal{S}} \leq \frac{2\sigma_g^2}{I^2} + \frac{2\sigma_{\boldsymbol{\alpha}}^2}{B},$$

*where $\sigma_g^2 = \frac{8C^2|\mathcal{V}|}{\tau^2\varepsilon^2}, \mathcal{S} = \{x_i, y_i\}_{i=1}^{B}$ and $\boldsymbol{\alpha}_{i,j} \geq \varepsilon > 0$.*

**Remark 2.** *This proposition demonstrates that the variance of the policy gradient is bounded under the assumptions provided. The bounds depend on the mini-batch size $B$ and the sampling times $I$, indicating that the variance can be controlled by choosing larger mini-batches and improving the gradient approximation technique.*

Based on these assumptions, we establish the following convergence result:

**Theorem 1.** *Under Assumption 1, 2, 3 and 4, let $\{\boldsymbol{\alpha}^t, \mathbf{z}^t\}_{t=0}^{T-1}$ be the sequence generated by running Algorithm 1, set $\eta_{\boldsymbol{\alpha}} \leq \frac{1}{2L_{\boldsymbol{\alpha}}}, \eta_{\mathbf{z}} \leq \frac{1}{40(d+4)L_{\mathbf{z}}}, I = \mathcal{O}\left(\frac{\sqrt{n\kappa}\sigma_g}{\epsilon}\right), B = \mathcal{O}\left(\max\{\frac{n\kappa\sigma_{\boldsymbol{\alpha}}^2}{\epsilon^2}, \frac{\kappa\sigma_{\mathbf{z}}^2}{\epsilon^2}\}\right), \mu = \mathcal{O}\left(\frac{1}{L_{max}}\sqrt{\frac{\epsilon^2}{d}}\right)$ and $T = \mathcal{O}\left(\frac{L_{max}}{\epsilon^2}\right)$, we have*

$$\frac{1}{T}\sum_{t=0}^{T-1}\mathbb{E}\left(\|\nabla_{\boldsymbol{\alpha}}\mathcal{L}(\boldsymbol{\alpha}^t, \mathbf{z}^t)\|^2 + \|\nabla_z\mathcal{L}(\boldsymbol{\alpha}^{t+1}, \mathbf{z}^t)\|^2\right) \leq \mathcal{O}(\epsilon^2), \tag{10}$$

*where $L_{max} = \max\{L_{\boldsymbol{\alpha}}, (d+4)L_{\mathbf{z}}\}, L_{min} = \min\{L_{\boldsymbol{\alpha}}, (d+4)L_{\mathbf{z}}\}$, and $\kappa = L_{max}/L_{min}$.*

**Remark 3.** *The above theorem establishes that ZO-PoG achieves an $\epsilon$-stationary point in expectation with total query complexity (forward passes of the PTMs) $T \cdot I = \mathcal{O}\left(\frac{\sqrt{n\kappa}}{\epsilon^3}\right)$.*

## 5 EXPERIMENTS

### 5.1 EXPERIMENT SETUPS

**Datasets.** For performance evaluation, we chose 5 commonly utilized datasets from the GLUE benchmark (Wang et al., 2018): CoLA (Warstadt et al., 2018), MNLI (Williams et al., 2017), QNLI (Wang et al., 2019), SNLI (Bowman et al., 2015), and WNLI (Levesque et al., 2012). These datasets encompass various typical language understanding tasks such as natural language inference.

**Baselines.** In our comparative analysis, we evaluate our proposed method against the following black-box prompt learning techniques under the same experimental conditions: **Manual Prompt** (MP) executes zero-shot evaluations on LLMs with human-crafted templates, providing initial performance metrics. **BBT** (Sun et al., 2022b) optimizes continuous prompts within a random low-dimensional subspace utilizing the Covariance Matrix Adaptation Evolution Strategy. **BDPL** (Diao et al., 2022) is a policy gradient-based black-box discrete prompt learning method, which is mentioned in section 3.2. **SSPT** (Zhang et al., 2024) builds upon the BBT optimization paradigm by employing subspace learning and selection to identify the optimal ultra-low-dimensional subspace, replacing the random subspace.

**Implementation Details.** The experiments are executed on a cluster of NVIDIA A40 GPUs. We employ RoBERTa-large (Liu et al., 2019), GPT2-XL (Radford et al., 2019), and Llama3 (AI@Meta, 2024) as our backbone models, and all pre-trained weights are sourced directly from HuggingFace. All experiments are performed under the few-shot learning setting. We assemble the training and development sets by randomly selecting $m$ instances for each class from the original training data. Comprehensive details of the input templates and hyperparameters used in our experiments can be found in Appendix B.

Table 1: Comparison of test accuracy (mean±std) on RoBERTa-Large in 16-shot (per class) setting. The best results are underlined. The **bold** represents the optimal result in the prompt learning approach. **Len** represents the prompt length.

| Len | Method | CoLA | MNLI | QNLI | SNLI | WNLI |
|-----|--------|------|------|------|------|------|
| - | MP | 48.51 | 41.61 | 51.91 | 38.00 | 42.25 |
| 20 | BBT | 56.18±3.34 | 40.55±0.08 | 52.97±1.58 | 37.39±0.33 | 50.23±6.95 |
| | BDPL | 38.83±1.00 | 43.22±1.63 | 55.25±2.39 | 38.04±1.30 | 39.91±2.93 |
| | SSPT | 44.65±7.52 | 41.72±0.40 | 53.10±2.23 | 36.84±2.99 | 46.01±3.54 |
| | Ours | **56.79±8.28** | **43.29±0.70** | **55.44±0.98** | **39.12±0.29** | **53.99±0.81** |
| 50 | BBT | 49.76±4.51 | 41.95±0.08 | 52.83±0.86 | 38.80±0.35 | 44.60±6.95 |
| | BDPL | 42.73±4.64 | 40.98±1.13 | 54.74±1.00 | 38.48±0.69 | 43.19±0.81 |
| | SSPT | 49.60±1.20 | 42.44±0.44 | 53.74±0.14 | 41.52±2.05 | 51.17±4.07 |
| | Ours | **56.18±7.59** | **42.53±0.81** | **55.33±0.32** | **41.99±0.60** | **56.34±1.41** |

Table 2: Comparison of test accuracy (mean±std) on GPT2-XL in 16-shot (per class) setting. The best results are underlined. The **bold** represents the optimal result in the prompt learning approach. **Len** represents the prompt length.

| Len | Method | CoLA | MNLI | QNLI | SNLI | WNLI |
|-----|--------|------|------|------|------|------|
| - | MP | 67.88 | 37.25 | 50.30 | 35.16 | 47.89 |
| 20 | BBT | 30.97±0.10 | 33.81±0.30 | 50.73±0.30 | 33.25±0.09 | 55.40±1.63 |
| | BDPL | 45.29±7.73 | 38.18±0.43 | 51.96±0.81 | 35.61±1.08 | 42.25±2.82 |
| | SSPT | 49.89±3.54 | 37.93±2.16 | 50.97±0.69 | 36.25±0.44 | 48.83±1.63 |
| | Ours | **53.69±3.74** | **38.78±1.53** | **52.21±1.93** | **37.17±1.75** | **56.81±0.81** |
| 50 | BBT | 31.83±0.10 | 33.62±0.61 | 50.72±0.17 | 33.27±0.12 | 55.40±1.63 |
| | BDPL | 49.15±11.82 | 37.23±0.35 | 51.75±1.06 | 35.17±0.93 | 41.78±4.53 |
| | SSPT | 53.18±3.29 | 37.81±2.02 | 51.17±1.24 | 37.84±0.34 | 53.52±2.82 |
| | Ours | **58.87±2.55** | **39.58±0.60** | **51.85±0.49** | **37.88±0.35** | **56.34±2.44** |

Table 3: Comparison of test accuracy (mean±std) on Llama3 in 16-shot (per class) setting. The best results are underlined. The **bold** represents the optimal result in the prompt learning approach. **Len** represents the prompt length.

| Len | Method | CoLA | MNLI | QNLI | SNLI | WNLI |
|---|---|---|---|---|---|---|
| - | MP | 60.50 | 33.98 | 52.65 | 38.01 | 57.75 |
| 20 | BBT | 45.89±7.32 | 33.30±0.65 | 49.16±0.30 | 31.88±1.08 | 47.42±5.33 |
| | BDPL | 45.83±7.12 | 37.33±3.13 | 55.11±1.21 | 34.45±1.16 | 54.93±2.44 |
| | SSPT | 41.87±2.13 | 38.25±3.44 | 54.24±1.09 | 36.70±3.00 | 53.52±2.82 |
| | Ours | **48.13±5.93** | **38.84±2.46** | **55.90±2.89** | **38.26±2.85** | **59.15±3.73** |
| 50 | BBT | 44.84±4.61 | 33.70±0.24 | 50.29±0.33 | 33.20±0.39 | 51.64±3.25 |
| | BDPL | 39.98±1.26 | 37.06±1.65 | 53.80±1.76 | 37.32±3.17 | 57.28±0.81 |
| | SSPT | 44.97±9.63 | 36.14±1.99 | 52.20±2.66 | 34.01±1.07 | 55.40±1.63 |
| | Ours | **46.50±6.98** | **37.19±0.77** | **55.60±1.22** | **38.12±0.86** | **58.22±2.15** |

## 5.2 MAIN RESULTS

We perform experiments with different prompt lengths and report the mean test accuracy over 3 random seeds. The comparative results for each of the 3 backbone models are detailed in Tables 1, 2, and 3, respectively. Notably, the proposed collaborative optimization method outperforms other prompt learning methods across all datasets. For instance, when employing the RoBERTa-large model on WNLI with a prompt length of 50, ZO-PoG achieves an improvement of 5.17% over the best baseline result. The experimental results confirm the effectiveness of jointly optimizing both the discrete initialization prompt and the continuous prompt. When using GPT2-XL and Llama3 as backbone models on the CoLA dataset, all prompt learning methods perform worse than manual prompts. This can be attributed to the CoLA task's focus on grammatical acceptability, which aligns better with the pretraining objectives of encoder-only models like RoBERTa. Decoder-only models like GPT2-XL and Llama3, pretrained for generative tasks, are less sensitive to grammatical correctness unless explicitly encoded in the prompt. Additionally, learned prompts, such as those from BDPL, may include grammatically incorrect tokens, further impairing the performance of decoder-only models. In contrast, manual prompts, designed with grammatical correctness, provide clearer guidance for these models, explaining their superior performance on CoLA. Additionally, the results for prompt lengths of 20 and 50 do not demonstrate a strict correlation between prompt length and performance. We believe that shorter prompt lengths may constrain representational capacity, whereas longer prompt lengths increase the number of parameters, making optimization more challenging. We conducted an ablation study on the effect of prompt length in Appendix C.

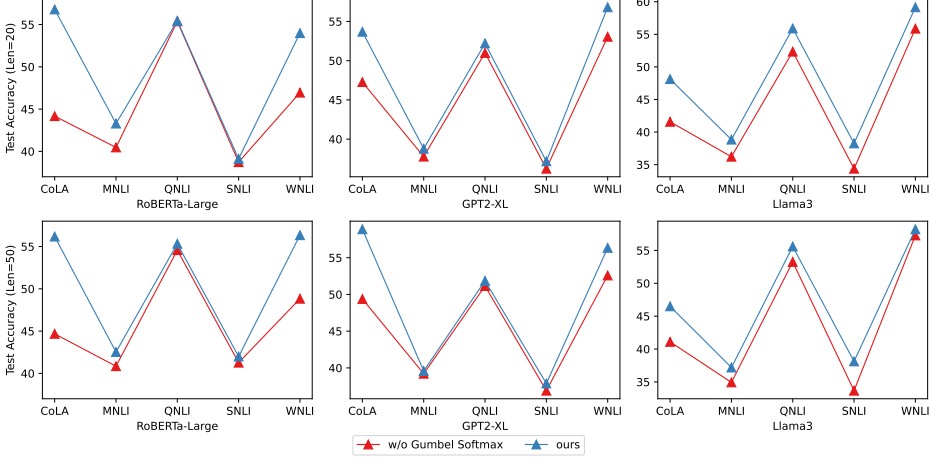

Figure 2: Ablations of the Gumbel-Softmax trick on RoBERTa-Large, GPT2-XL and Llama3 with the prompt lengths of 20 (top) and 50 (bottom).

## 5.3 ABLATION STUDY

**Effect of Gumbel-Softmax Trick.** We removed the Gumbel-Softmax trick from ZO-PoG and replaced it with the policy gradient estimation method used in (Diao et al., 2022). The comparative results are presented in Figure 2. The results indicate that the Gumbel-Softmax trick has a positive impact on the overall optimization in ZO-PoG.

**Effect of Optimization Component.** We separately removed the discrete prompt optimization component (w/o Policy Gradient) and the continuous prompt optimization component (w/o Zeroth Order) from ZO-PoG. The experimental results for the prompt lengths of 20 and 50 are presented in Figure 3. It can be observed that ZO-PoG consistently maintains the best performance across all backbone models. This demonstrates that ZO-PoG is capable of effectively combining both prompt learning approaches, significantly enhancing the performance of LLMs on downstream tasks.

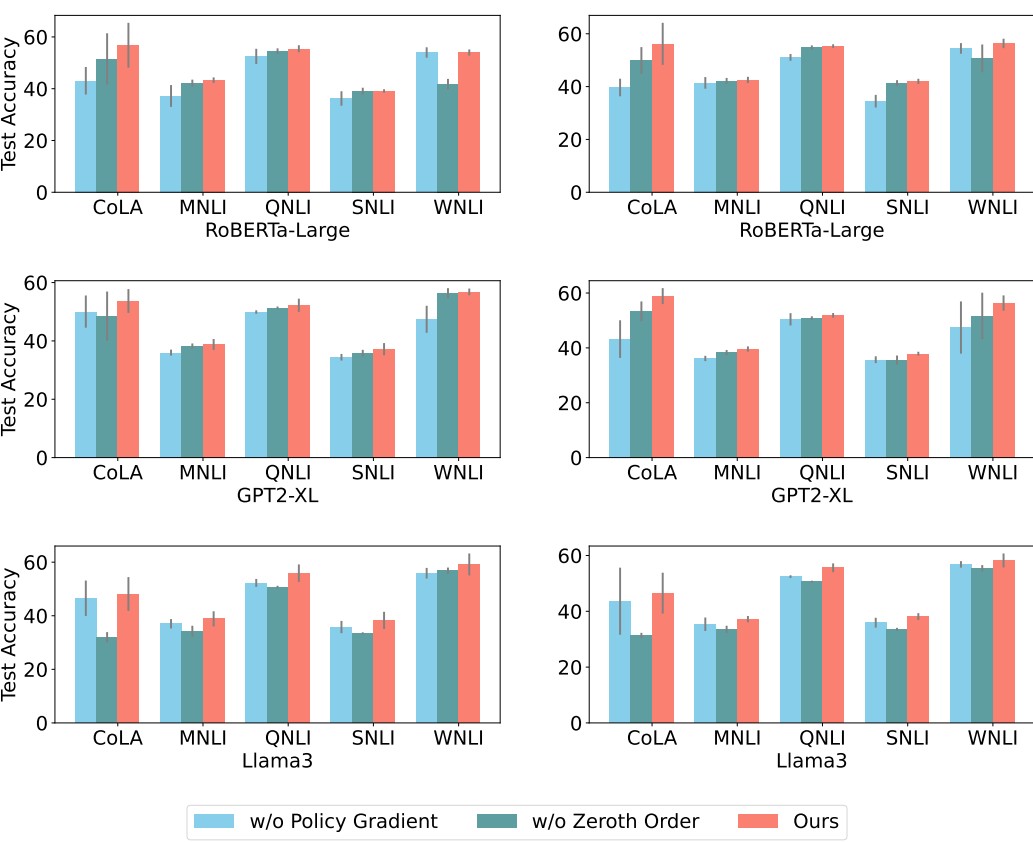

Figure 3: Ablations of the collaborative optimization components on RoBERTa-Large, GPT2-XL and Llama3 with the prompt lengths of 20 (left) and 50 (right).

## 6 CONCLUSION

In this paper, we propose a novel black-box prompt tuning framework for PTMs named **Z**eroth-**O**rder and **Po**licy **G**radient (ZO-PoG) method, which optimizes both discrete and continuous prompts in the black-box setting. During the tuning process, we alternately optimize the **discrete** prompts using the policy gradient in its parameters space and **continuous** prompts using the zeroth-order gradient in the parameter's low-intrinsic space. Our experiments show that ZO-PoG significantly improves the Large language model's performance in various downstream tasks while reducing the computational expense in the Black-box setting. Moreover, our convergence analysis proves the effectiveness of our method.

ACKNOWLEDGMENTS

This work was supported in part by the National Key R&D Program of China under Grant (No.2023YFF0905400), National Natural Science Foundation of China through grants (No.U2341229, No.62076138)

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

# APPENDIX

# A    CONVERGENCE RATE ANALYSIS

## A.1    AUXILIARY LEMMAS

**Lemma 1.** *(Notation in Nesterov & Spokoiny (2017)) Denote: $f \in C^{1,1}(\mathbb{R}^d)$, if:*
$$\|\nabla f(x) - \nabla f(y)\| \leq L\|y - x\|,$$
*where $x, y \in \mathbb{R}^d$. This condition is equivalent to the following inequality:*
$$|f(y) - f(x) - \langle \nabla f(x), y - x \rangle| \leq \frac{L^2}{2}\|y - x\|^2, \quad x, y \in \mathbb{R}^d. \tag{11}$$

**Lemma 2.** *(Lemma 1 in Nesterov & Spokoiny (2017)) We often need bounds for the moments:*
$$M_p := \frac{1}{\kappa} \int_{\mathbb{R}^d} \|u\|^p e^{-\frac{1}{2}\|u\|^2} \mathrm{d}u,$$
*where $\kappa = \int_{\mathbb{R}^d} e^{-\frac{1}{2}\|u\|^2} \mathrm{d}u$. For $p \in [0, 2]$, $d$ represents the dimensional of the space, then we have:*
$$M_p \leq d^{p/2}.$$
*If $p \geq 2$, then we have two-side bounds:*
$$d^{p/2} \leq M_p \leq (p + d)^{p/2}. \tag{12}$$

**Lemma 3.** *(Theorem 3 in Nesterov & Spokoiny (2017)) If $f$ is differentiable at $x$, then*
$$\mathbb{E}_u\left(\|g_0(x)\|^2\right) \leq (d + 4)\|\nabla f_0(x)\|^2. \tag{13}$$
*Denote: $g_0(x) := f'(x, u) \cdot u$ where $f'(x, u)$ is a directional derivative of function $f(x)$ along $u \in \mathbb{R}^n$, and $\nabla f_0(x) := \frac{1}{\kappa} \int_{\mathbb{R}^d} \langle \nabla f(x), u \rangle e^{-\frac{1}{2}\|u\|^2} u \mathrm{d}u$.*

**Lemma 4.** *$f \in C^{1,1}(\mathbb{R}^d)$, which means that $\|\nabla f(x) - \nabla f(y)\| \leq L\|x - y\|, x, y \in \mathbb{R}^d$. Then, let random vectors $u_i \in \mathbb{R}^d$ have stand Gaussian distribution $\mathcal{N}(0, \mathbf{I}_d)$ and return*
$$\hat{g}_\mu(x) := \frac{f(x + \mu u_i) - f(x - \mu u_i)}{2\mu} u_i.$$
*Finally, we will have a conclusion:*
$$E_u\left(\|\hat{g}_\mu(x)\|^2\right) \leq \frac{\mu^2}{2}L^2(d+6)^3 + 2(d+4)\|\nabla f(x)\|^2. \tag{14}$$

*Proof.* For the symmetric oracle $\hat{g}_\mu$, we have
$$|f(x + \mu u) - f(x - \mu u)|^2 = |f(x + \mu u) - f(x) - \langle \nabla f(x), \mu u \rangle - (f(x - \mu u) - f(x) + \langle \nabla f(x), \mu u \rangle)$$
$$+ 2\langle \nabla f(x), \mu u \rangle|^2$$
$$\overset{(a)}{\leq} 2|f(x + \mu u) - f(x) - \langle \nabla f(x), \mu u \rangle + \langle \nabla f(x), \mu u \rangle|^2$$
$$+ 2|f(x - \mu u) - f(x) + \langle \nabla f(x), -\mu u \rangle + \langle \nabla f(x), \mu u \rangle|^2$$
$$\overset{(11)}{\leq} 2\left|\frac{\mu^2}{2}L\|u\|^2 + \langle \nabla f(x), \mu u \rangle\right|^2 + 2\left|\frac{\mu^2}{2}L\|u\|^2 + \langle \nabla f(x), \mu u \rangle\right|^2$$
$$= 4\left|\frac{\mu^2}{2}L\|u\|^2 + \mu\langle \nabla f(x), u \rangle\right|^2.$$

Then, take the expectation with respect to $u$ following the Gaussian distribution:
$$E_u\left(\|\hat{g}_\mu(x)\|^2\right) = \frac{1}{4\mu^2}E_u\left(|f(x + \mu u) - f(x - \mu u)|^2\|u\|^2\right)$$
$$\overset{(a)}{\leq} \frac{2}{\mu^2}\left[E_u\left(\frac{\mu^4}{4}L^2\|u\|^6\right) + E_u\left(\mu^2\langle \nabla f(u), u \rangle^2\|u\|^2\right)\right]$$
$$\overset{(12)(13)}{\leq} \frac{\mu^2}{2}L^2(d+6)^3 + 2(d+4)\|\nabla f(x)\|^2,$$
where inequality $(a)$ holds because $\mathbb{E}\|a + b\|^2 \leq 2\mathbb{E}\|a\|^2 + 2\mathbb{E}\|b\|^2$. □

**Lemma 5.** *(Theorem 1 and Lemma 3 in Nesterov & Spokoiny (2017)) If $f \in C^{1,1}(\mathbb{R}^d)$ with constant $L$, and $x \in \mathbb{R}^d$ then for the bias with respect to the zeroth-order function:*

$$|f_\mu(x) - f(x)| \leq \frac{\mu^2}{2} Ld. \tag{15}$$

*For the bias with respect to the zeroth-order gradient:*

$$\|\nabla f_\mu(x) - \nabla f(x)\| \leq \frac{\mu}{2} L(d+3)^{3/2}, \tag{16}$$

*where:*

$$f_\mu(x) = \frac{1}{\kappa} \int_{\mathbb{R}^d} f(x + \mu u) e^{-\frac{1}{2}\|u\|^2} \mathrm{d}u;$$

$$\nabla f_\mu(x) = \frac{1}{\kappa} \int_{\mathbb{R}^d} \frac{f(x+\mu u) - f(x-\mu u)}{2\mu} e^{-\frac{1}{2}\|u\|^2} u \mathrm{d}u.$$

**Proposition 2.** *Now, let's analyze the boundedness of the zeroth-order gradient's variance. For any $x \in \mathbb{R}^d$, we have:*

$$\mathbb{E}\left\|\frac{1}{B}\sum_{i\in S} \hat{\nabla} f_i(x) - \hat{\nabla} f(x)\right\|^2 \leq \frac{4(d+4)}{B}\|\nabla f(x)\|^2 + \frac{4(d+4)\sigma_{\mathbf{z}}^2}{B} + \frac{\mu^2}{8B}L^2(d+6)^3, \tag{17}$$

*where $\hat{\nabla} f_i(x)$ and $\hat{\nabla} f(x)$ are the estimated gradient by the zeroth-order gradients. For $\hat{\nabla} f_i(x)$, we will sample one random disturbance $u_i$ per sample; $u_i$ follows Gaussian distribution. Then denote: $\sigma_{\mathbf{z}}^2 := \mathbb{E}_i \|\nabla f_i(x) - \nabla f(x)\|^2$ to represent the sample variance of the data sets.*

*Proof.* Let $\mathbb{I}_S(\cdot)$ be a indicator function, $\mathbb{I}_S(i) = \begin{cases} 1, & \text{if } i \in S \\ 0, & \text{if } i \notin S \end{cases}$, where $S$ represents the mini-batch set and denote $\mathbb{I}_i := \mathbb{I}_S(i)$. Then we have $\mathbb{E}_i\left(\mathbb{I}_i^2\right) = \frac{B}{n}$ and $\mathbb{E}_i\left(\mathbb{I}_i\mathbb{I}_j\right) = \frac{\binom{B}{2}}{\binom{n}{2}} = \frac{B(B-1)}{n(n-1)}$, when $i \neq j$. Let $z_i = \hat{\nabla} f_i(x) - \hat{\nabla} f(x)$:

$$\mathbb{E}\left\|\frac{1}{B}\sum_{i\in S} \hat{\nabla} f_i(x) - \hat{\nabla} f(x)\right\|^2 = \mathbb{E}\left\|\frac{1}{B}\sum_{i\in S} z_i\right\|^2 = \frac{1}{B^2}\mathbb{E}\left\|\sum_{i=1}^n z_i \mathbb{I}_i\right\|^2$$

$$= \frac{1}{|S|^2}\left(\sum_{i=1}^n \mathbb{E}\mathbb{I}_i^2 \|z_i\|^2 + \sum_{i\neq j} \mathbb{E}\mathbb{I}_i\mathbb{I}_j \langle z_i, z_j\rangle\right) = \frac{1}{B^2}\mathbb{E}_u\left(\frac{B}{n}\sum_{i=1}^n \|z_i\|^2 + \frac{B(B-1)}{n(n-1)}\sum_{i\neq j}\langle z_i, z_j\rangle\right)$$

$$= \frac{1}{B^2}\mathbb{E}_u\left(\left(\frac{B}{n} - \frac{B(B-1)}{n(n-1)}\right)\sum_{i=1}^n \|z_i\|^2 + \frac{B(B-1)}{n(n-1)}\left\|\sum_{i=1}^n z_i\right\|^2\right)$$

$$= \mathbb{E}_u \frac{n-B}{n(n-1)B}\sum_{i=1}^n \|z_i\|^2 \overset{(a)}{\leq} \frac{1}{B}\mathbb{E}_u \frac{1}{n}\sum_{i=1}^n \|z_i\|^2 = \frac{1}{B}\mathbb{E}_u\mathbb{E}_i\left\|\hat{\nabla} f_i(x) - \hat{\nabla} f(x)\right\|^2$$

$$\leq \frac{1}{B}\mathbb{E}_u\mathbb{E}_i\left\|\hat{\nabla} f_i(x)\right\|^2$$

$$\overset{(14)}{\leq} \frac{1}{B}\mathbb{E}_i\left(\frac{\mu^2}{2}L^2(d+6)^3 + 2(d+4)\|\nabla f(x)\|^2\right)$$

$$\overset{(b)}{\leq} \frac{\mu^2}{2B}L^2(d+6)^3 + \frac{4(d+4)}{B}\mathbb{E}_i\left(\|\nabla f_i(x) - \nabla f(x)\|^2 + \|\nabla f_i(x)\|^2\right)$$

$$= \frac{4(d+4)}{B}\|\nabla f(x)\|^2 + \frac{4(d+4)\sigma_{\mathbf{z}}^2}{B} + \frac{\mu^2}{2B}L^2(d+6)^3,$$

where inequality $(a)$ holds because $\frac{n-B}{n(n-1)B} \leq \frac{1}{B}\frac{n-1}{n(n-1)} \leq \frac{1}{B}\frac{1}{n}$; inequality $(b)$ holds because $\mathbb{E}\|a+b\|^2 \leq 2\mathbb{E}\|a\|^2 + 2\mathbb{E}\|b\|^2$.

$\square$

## A.2 PROOF OF PROPOSITION 1

*Proof.* The variance between the policy gradient and true gradient can divided into two parts, The first variance part comes from the sampling process with fixed mini-batch $S$ and fixed $P(T)$, and the second variance part comes from stochastic mini-batch $S = \{x_i, y_i\}_{i=1}^{B}$. (Notice that the mini-batch we use in both zeroth-order and policy gradient methods are the same)

$$\mathbb{E}_S \mathbb{E}_{T \sim P(T)} \|g_{\boldsymbol{\alpha}_i}^{vr} - \nabla_{\boldsymbol{\alpha}_i} \mathcal{L}(\boldsymbol{\alpha}, \mathbf{z})\|^2 \tag{18}$$

$$\leq 2 \underbrace{\mathbb{E}_{T \sim P(T)} \|g_{\boldsymbol{\alpha}_i}^{vr} - \nabla_{\boldsymbol{\alpha}_i} \mathcal{L}_S(\boldsymbol{\alpha}, \mathbf{z})\|^2}_{\text{Where } S \text{ is fixed}} + 2 \underbrace{\mathbb{E}_S \|\nabla_{\boldsymbol{\alpha}_i} \mathcal{L}_S(\boldsymbol{\alpha}, \mathbf{z}) - \nabla_{\boldsymbol{\alpha}_i} \mathcal{L}(\boldsymbol{\alpha}, \mathbf{z})\|^2}_{\text{Variance from mini-batch } S}. \tag{19}$$

Then we analyze the first part of the variance, where $S$ is fixed.

$$\text{Var}(g_{\boldsymbol{\alpha}_i}^{vr}) = \text{Var}\left( \frac{1}{I-1} \sum_{k=1}^{I} \nabla_{\boldsymbol{\alpha}_i} \log P(t_i^{(k)}) \left( \mathcal{L}_S(\boldsymbol{\alpha}^{(k)}, \mathbf{z}) - \mathcal{L}_{\text{avg}} \right) \right)$$

$$= \text{Var}\left[ \frac{1}{I} \sum_{k=1}^{I} \frac{1}{I-1} \sum_{j \neq k} \left( \mathcal{L}_S(\boldsymbol{\alpha}^{(k)}, \mathbf{z}) - \mathcal{L}_S(\boldsymbol{\alpha}^{(j)}, \mathbf{z}) \right) \nabla_{\boldsymbol{\alpha}_i} \log P(t_i^{(k)}) \right]$$

$$= \mathbb{E}\left\| \frac{1}{I} \sum_{k=1}^{I} \frac{1}{I-1} \sum_{j \neq k} \left( \mathcal{L}_S(\boldsymbol{\alpha}^{(k)}, \mathbf{z}) - \mathcal{L}_S(\boldsymbol{\alpha}^{(j)}, \mathbf{z}) \right) \nabla_{\boldsymbol{\alpha}_i} \log P(t_i^{(k)}) - \nabla_{\boldsymbol{\alpha}_i} \mathcal{L}_S(\boldsymbol{\alpha}, \mathbf{z}) \right\|^2$$

$$= \mathbb{E}\left\| \frac{1}{I} \sum_{k=1}^{I} \left[ \frac{1}{I-1} \sum_{j \neq k} \left( \mathcal{L}_S(\boldsymbol{\alpha}^{(k)}, \mathbf{z}) - \mathcal{L}_S(\boldsymbol{\alpha}^{(j)}, \mathbf{z}) \right) \nabla_{\boldsymbol{\alpha}_i} \log P(t_i^{(k)}) - \nabla_{\boldsymbol{\alpha}_i} \mathcal{L}_S(\boldsymbol{\alpha}, \mathbf{z}) \right] \right\|^2$$

$$\leq \frac{1}{I^2} \mathbb{E} \sum_{k=1}^{I} \left\| \frac{1}{I-1} \sum_{j \neq k} \left( \mathcal{L}_S(\boldsymbol{\alpha}^{(k)}, \mathbf{z}) - \mathcal{L}_S(\boldsymbol{\alpha}^{(j)}, \mathbf{z}) \right) \nabla_{\boldsymbol{\alpha}_i} \log P(t_i^{(k)}) - \nabla_{\boldsymbol{\alpha}_i} \mathcal{L}_S(\boldsymbol{\alpha}, \mathbf{z}) \right\|^2$$

$$= \frac{1}{I} \mathbb{E}_{\boldsymbol{\alpha}^{(k)}} \left\| \frac{1}{I-1} \sum_{j \neq k} \left[ \left( \mathcal{L}_S(\boldsymbol{\alpha}^{(k)}, \mathbf{z}) - \mathcal{L}_S(\boldsymbol{\alpha}^{(j)}, \mathbf{z}) \right) \nabla_{\boldsymbol{\alpha}_i} \log P(t_i^{(k)}) - \nabla_{\boldsymbol{\alpha}_i} \mathcal{L}_S(\boldsymbol{\alpha}, \mathbf{z}) \right] \right\|^2$$

$$\overset{(c)}{\leq} \frac{1}{I(I-1)} \mathbb{E}_{\boldsymbol{\alpha}^{(k)}, \boldsymbol{\alpha}^{(j)}, k \neq j} \left\| \left( \mathcal{L}_S(\boldsymbol{\alpha}^{(k)}, \mathbf{z}) - \mathcal{L}_S(\boldsymbol{\alpha}^{(j)}, \mathbf{z}) \right) \nabla_{\boldsymbol{\alpha}_i} \log P(t_i^{(k)}) \right\|^2$$

$$\overset{(d)}{\leq} \frac{4C^2}{I(I-1)} \mathbb{E}_{\boldsymbol{\alpha}^{(k)}, \boldsymbol{\alpha}^{(j)}, k \neq j} \left\| \nabla_{\boldsymbol{\alpha}_i} \log P(t_i^{(k)}) \right\|^2$$

$$\overset{(e)}{\leq} \frac{4C^2 |\mathcal{V}|}{I(I-1)\tau^2 \varepsilon^2} \overset{(f)}{\leq} \frac{8C^2 |\mathcal{V}|}{\tau^2 \varepsilon^2 I^2}.$$

Above, inequality $(c)$ holds because $\mathbb{E}\|a - b\|^2 \leq \mathbb{E}\|a\|^2$. Inequality $(d)$ holds because Assumption 3, there exists a positive constant $C < \infty$ such that $|\mathcal{L}_S(\boldsymbol{\alpha}, \mathbf{z})| < C, \forall \boldsymbol{\alpha}, \mathbf{z}$. Inequality $(e)$ holds because the $j$-th component of $\nabla_{\boldsymbol{\alpha}_i} \log P(t_i)$ could be solved explicitly by:

$$\nabla_{\boldsymbol{\alpha}_{i,j}} \log P(t_i | \boldsymbol{\alpha}_i) = \nabla_{\boldsymbol{\alpha}_{i,j}} \log p_{i,j_i} = \begin{cases} \nabla_{\boldsymbol{\alpha}_{i,j}} \log P(t_i) = (1 - p_{i,j_i}) \dfrac{1}{\tau \boldsymbol{\alpha}_{i,j_i}}, & j = j_i; \\ \nabla_{\boldsymbol{\alpha}_{i,j}} \log P(t_i) = -p_{i,j} \dfrac{1}{\tau \boldsymbol{\alpha}_{i,j}}, & j \neq j_i. \end{cases}$$

Then we have

$$\left\| \nabla_{\boldsymbol{\alpha}_i} \log P(t_i^{(k)}) \right\|^2 \leq |\mathcal{V}| \max \left\{ (1 - p_{i,j_i}) \frac{1}{\tau \boldsymbol{\alpha}_{i,j_i}}, p_{i,j} \frac{1}{\tau \boldsymbol{\alpha}_{i,j}} \right\}^2 \leq \frac{|\mathcal{V}|}{\tau^2 \varepsilon^2}.$$

The last inequality $(f)$ holds when $I \geq 2$, where $|\mathcal{V}|$ means the total number of the tokens in the vocabulary list.

Denote $\sigma_g^2 = \dfrac{8C^2|\mathcal{V}|}{\tau^2 \varepsilon^2}$, then we have:

$$\mathbb{E}_{\boldsymbol{\alpha} \sim P(T)} \|\nabla g_{\boldsymbol{\alpha}_i}^{vr} - \nabla \mathcal{L}_{\mathcal{S}}(\boldsymbol{\alpha}, \mathbf{z})\|^2 \leq \frac{\sigma_g^2}{I^2}.$$

On the other hand, under assumption 2, it is easy to get $\mathbb{E}_{\mathcal{S}} \|\nabla_{\boldsymbol{\alpha}_i} \mathcal{L}_{\mathcal{S}}(\boldsymbol{\alpha}, \mathbf{z}) - \nabla_{\boldsymbol{\alpha}_i} \mathcal{L}(\boldsymbol{\alpha}, \mathbf{z})\|^2 \leq \dfrac{\sigma_{\boldsymbol{\alpha}}^2}{B}$. This completes the proof of Proposition 1.

$\square$

## A.3 PROOF OF THE THEOREM 1

**Notation recap:**

- $\mathcal{L}(\boldsymbol{\alpha}, \mathbf{z})$ is an abbreviation of the loss function. We utilize the cross-entropy loss function during the experiment.

- $\mathcal{L}_\mu(\boldsymbol{\alpha}, \mathbf{z})$ represents Gaussian smoothing of the original loss function:

$$\mathcal{L}_\mu(\boldsymbol{\alpha}, \mathbf{z}) = \frac{1}{\kappa} \int_{\mathbb{R}^d} \mathcal{L}(\boldsymbol{\alpha}, \mathbf{z} + \mu u) e^{-\frac{1}{2}\|u\|^2} \mathrm{d}u,$$

  where $\mu$ is hyper-parameter and $\kappa = \int_{\mathbb{R}^d} e^{-\frac{1}{2}\|u\|^2} \mathrm{d}u$.

- $\nabla_{\boldsymbol{\alpha}} \mathcal{L}(\cdot)$ means that the block-wise true gradient with respect to $\boldsymbol{\alpha}$ and, $\nabla_{\mathbf{z}} \mathcal{L}(\cdot)$, $\nabla_{\mathbf{z}} \mathcal{L}_\mu(\cdot)$ represents that the block-wise true gradient with and zeroth-order gradient with respect to $\mathbf{z}$. Where $\boldsymbol{\alpha}$ is the parameter that can create discrete prompts distribution, optimized by the policy gradient algorithm and $\mathbf{z}$ is the continuous prompt in the low-dimensional space optimized by the zeroth-order gradient.

- $g_{\mathbf{z}}$ means the estimated gradient with a random mini-batch, and the $\hat{\nabla}_{\mathbf{z}} \mathcal{L}(\cdot)$ represents the estimated gradient calculated by a total dataset.

- $g_{\boldsymbol{\alpha}}^{vr}$ represents the policy gradient in the parameter space.

- $\eta_{\mathbf{z}}$ and $\eta_{\boldsymbol{\alpha}}$ represent the learning rates with respect to $\nabla_{\mathbf{z}} \mathcal{L}(\cdot)$ and $\nabla_{\boldsymbol{\alpha}} \mathcal{L}(\cdot)$ respectively.

- $d$ represents the dimension of the parameter space, and $n$ represents the length of the initialized discrete prompts. And each token is independent with others, so we can add their variances together directly.

- $L_{\mathbf{z}}$, $L_{\boldsymbol{\alpha}}$ represent the Lipschitz constants with respect to L-Smooth functions $\mathcal{L}_\mu(\cdot)$ and $\mathcal{L}_{\boldsymbol{\alpha}}(\cdot)$ respectively.

- $B$ is the size of the mini-batch, represented as $S_t = \{X_t, Y_t\} = \{x_i, y_i\}_{i=1}^B$, and $I$ is the sampling times in the policy gradient every iteration.

- For convenience, let $\mathcal{L}(f(\mathbf{A}\mathbf{z} + \mathbf{p}_0; X), Y) := \mathcal{L}(\boldsymbol{\alpha}, \mathbf{z})$, where $f$ represents the Black-box API, total dataset $\mathcal{D} = \{X, Y\}$.

- $\sigma_{\mathbf{z}}$ is the variance from SGD during continuous prompt optimization, $\sigma_g$ is the variance induced by policy gradient, and $\sigma_{\boldsymbol{\alpha}}$ is the policy gradient variance from mini-batch.

*Proof.* According to the lemma 5, the value of the zeroth-order function (Gaussian smooth: Origin function with a Gaussian kernel) has a bias compared with the true function value:

$$|\mathcal{L}_\mu(\boldsymbol{\alpha}, \mathbf{z}) - \mathcal{L}(\boldsymbol{\alpha}, \mathbf{z})| \leq \frac{\mu^2}{2} L_{\mathbf{z}} d. \tag{20}$$

We can also know that $\mathcal{L}_\mu(\boldsymbol{\alpha}, \mathbf{z})$ with respect to $\mathbf{z}$ is a $L_{\mathbf{z}}$ Lipschitz smooth function and $\mathcal{L}(\boldsymbol{\alpha}, \mathbf{z})$ with respect to $\boldsymbol{\alpha}$, is a $L_{\boldsymbol{\alpha}}$ Lipschitz smooth function from the Assumption 1.

According to Eq.(12) in Nesterov & Spokoiny (2017), we can now that $\mathcal{L}_\mu$ is Lipschitz smooth for $\forall \mathbf{z}, \mathbf{z}', \boldsymbol{\alpha}$:

$$\|\nabla_{\mathbf{z}} \mathcal{L}_\mu(\boldsymbol{\alpha}, \mathbf{z}) - \nabla_{\mathbf{z}} \mathcal{L}_\mu(\boldsymbol{\alpha}, \mathbf{z}')\| \leq L_{\mathbf{z}} \|\mathbf{z} - \mathbf{z}'\|, \tag{21}$$

where $\quad \nabla_{\mathbf{z}} \mathcal{L}_\mu(\boldsymbol{\alpha}, \mathbf{z}) \quad = \quad \frac{1}{\kappa} \int_{\mathbb{R}^d} \frac{\mathcal{L}(\boldsymbol{\alpha}, \mathbf{z} + \mu\mathbf{u}) - \mathcal{L}(\boldsymbol{\alpha}, \mathbf{z} - \mu\mathbf{u})}{2\mu} e^{-\frac{1}{2}\|\mathbf{u}\|^2} u \mathrm{d}u, \quad$ and $\quad \kappa \quad =$ $\int_{\mathbb{R}^d} e^{-\frac{1}{2}\|u\|^2} \mathrm{d}u.$

Firstly, because of the smoothness of the $\mathcal{L}(\cdot)$ with respect to $z$ from the inequality (7) and the bias from the zeroth-order gradient from the inequality (15):

$$\mathcal{L}(\boldsymbol{\alpha}^{t+1}, \mathbf{z}^{t+1}) - \mathcal{L}(\boldsymbol{\alpha}^{t+1}, \mathbf{z}^t) \overset{(16)}{\leq} \left( \mathcal{L}_\mu(\boldsymbol{\alpha}^{t+1}, \mathbf{z}^{t+1}) + \frac{\mu^2}{2} L_{\mathbf{z}} d \right) - \left( \mathcal{L}_\mu(\boldsymbol{\alpha}^{t+1}, \mathbf{z}^t) - \frac{\mu^2}{2} L_{\mathbf{z}} d \right)$$

$$= \mathcal{L}_\mu(\boldsymbol{\alpha}^{t+1}, \mathbf{z}^{t+1}) - \mathcal{L}_\mu(\boldsymbol{\alpha}^{t+1}, \mathbf{z}^t) + \mu^2 L_{\mathbf{z}} d$$

$$\overset{(21)(11)}{\leq} \langle \nabla_{\mathbf{z}} \mathcal{L}_\mu(\boldsymbol{\alpha}^{t+1}, \mathbf{z}^t), \mathbf{z}^{t+1} - \mathbf{z}^t \rangle + \frac{L_{\mathbf{z}}}{2} \|\mathbf{z}^{t+1} - \mathbf{z}^t\|^2 + \mu^2 L_{\mathbf{z}} d.$$

Secondly, according to smoothness of the $\mathcal{L}(\cdot)$ with respect to $\boldsymbol{\alpha}$ according to the inequality (6) in the assumption 6, we use equation (11):

$$\mathcal{L}(\boldsymbol{\alpha}^{t+1}, \mathbf{z}^{t+1}) \leq \mathcal{L}(\boldsymbol{\alpha}^{t+1}, \mathbf{z}^t) + \langle \nabla_{\mathbf{z}} \mathcal{L}_\mu(\boldsymbol{\alpha}^{t+1}, \mathbf{z}^t), \mathbf{z}^{t+1} - \mathbf{z}^t \rangle + \frac{L_{\mathbf{z}}}{2} \|\mathbf{z}^{t+1} - \mathbf{z}^t\|^2 + \mu^2 L_{\mathbf{z}} d$$

$$\overset{(6)(11)}{\leq} \mathcal{L}(\boldsymbol{\alpha}^t, \mathbf{z}^t) + \langle \nabla_{\boldsymbol{\alpha}} \mathcal{L}(\boldsymbol{\alpha}^t, \mathbf{z}^t), \boldsymbol{\alpha}^{t+1} - \boldsymbol{\alpha}^t \rangle + \frac{L_{\boldsymbol{\alpha}}}{2} \|\boldsymbol{\alpha}^{t+1} - \boldsymbol{\alpha}^t\|^2$$

$$+ \langle \nabla_{\mathbf{z}} \mathcal{L}_\mu(\boldsymbol{\alpha}^{t+1}, \mathbf{z}^t), \mathbf{z}^{t+1} - \mathbf{z}^t \rangle + \frac{L_{\mathbf{z}}}{2} \|\mathbf{z}^{t+1} - \mathbf{z}^t\|^2 + \mu^2 L_{\mathbf{z}} d.$$

After rearranging the formulation above:

$$\mathbb{E} \mathcal{L}(\boldsymbol{\alpha}^{t+1}, \mathbf{z}^{t+1}) - \mathcal{L}(\boldsymbol{\alpha}^t, \mathbf{z}^t)$$

$$\leq \mathbb{E} \langle \nabla_{\boldsymbol{\alpha}} \mathcal{L}(\boldsymbol{\alpha}^t, \mathbf{z}^t), \boldsymbol{\alpha}^{t+1} - \boldsymbol{\alpha}^t \rangle + \frac{L_{\boldsymbol{\alpha}}}{2} \mathbb{E} \|\boldsymbol{\alpha}^{t+1} - \boldsymbol{\alpha}^t\|^2$$

$$+ \mathbb{E} \langle \nabla_{\mathbf{z}} \mathcal{L}_\mu(\boldsymbol{\alpha}^{t+1}, \mathbf{z}^t), \mathbf{z}^{t+1} - \mathbf{z}^t \rangle + \frac{L_{\mathbf{z}}}{2} \mathbb{E} \|\mathbf{z}^{t+1} - \mathbf{z}^t\|^2 + \mu^2 L_{\mathbf{z}} d$$

$$= - \eta_{\boldsymbol{\alpha}} \|\nabla_{\boldsymbol{\alpha}} \mathcal{L}(\boldsymbol{\alpha}^t, \mathbf{z}^t)\|^2 + \frac{\eta_{\boldsymbol{\alpha}}^2 L_{\boldsymbol{\alpha}}}{2} \mathbb{E} \|g_{\boldsymbol{\alpha}^t}^{vr}\|^2 - \eta_{\mathbf{z}} \|\nabla_{\mathbf{z}} \mathcal{L}_\mu(\boldsymbol{\alpha}^{t+1}, \mathbf{z}^t)\|^2 + \frac{\eta_{\mathbf{z}}^2 L_{\mathbf{z}}}{2} \mathbb{E} \|g_{\mathbf{z}^t}\|^2 + \mu^2 L_{\mathbf{z}} d$$

$$\overset{(g)}{\leq} - \eta_{\boldsymbol{\alpha}} \|\nabla_{\boldsymbol{\alpha}} \mathcal{L}(\boldsymbol{\alpha}^t, \mathbf{z}^t)\|^2 + \eta_{\boldsymbol{\alpha}}^2 L_{\boldsymbol{\alpha}} \mathbb{E}(\underbrace{\|g_{\boldsymbol{\alpha}^t}^{vr} - \nabla_{\boldsymbol{\alpha}} \mathcal{L}(\boldsymbol{\alpha}^t, \mathbf{z}^t)\|^2}_{Varicance\ (19)} + \|\nabla_{\boldsymbol{\alpha}} \mathcal{L}(\boldsymbol{\alpha}^t, \mathbf{z}^t)\|^2)$$

$$- \eta_{\mathbf{z}} \|\nabla_{\mathbf{z}} \mathcal{L}_\mu(\boldsymbol{\alpha}^{t+1}, \mathbf{z}^t)\|^2 + \frac{3\eta_{\mathbf{z}}^2 L_{\mathbf{z}}}{2} \mathbb{E}(\underbrace{\|g_{\mathbf{z}^t} - \hat{\nabla}_{\mathbf{z}} \mathcal{L}(\boldsymbol{\alpha}^{t+1}, \mathbf{z}^t)\|^2}_{Variance\ (17)} + \underbrace{\|\hat{\nabla}_{\mathbf{z}} \mathcal{L}(\boldsymbol{\alpha}^{t+1}, \mathbf{z}^t) - \nabla_{\mathbf{z}} \mathcal{L}_\mu(\boldsymbol{\alpha}^{t+1}, \mathbf{z}^t)\|^2}_{\mathbb{E}\|a - \mathbb{E}a\|^2 \leq \mathbb{E}\|a\|^2,\ then\ use\ (14)}$$

$$+ \|\nabla_{\mathbf{z}} \mathcal{L}_\mu(\boldsymbol{\alpha}^{t+1}, \mathbf{z}^t)\|^2) + \mu^2 L_{\mathbf{z}} d$$

$$\overset{(h)}{\leq} - (\eta_{\boldsymbol{\alpha}} - \eta_{\boldsymbol{\alpha}}^2 L_{\boldsymbol{\alpha}}) \mathbb{E} \|\nabla_{\boldsymbol{\alpha}} \mathcal{L}(\boldsymbol{\alpha}^t, \mathbf{z}^t)\|^2 + \eta_{\boldsymbol{\alpha}}^2 L_{\boldsymbol{\alpha}} n (\frac{2\sigma_g^2}{I^2} + \frac{2\sigma_{\boldsymbol{\alpha}}^2}{B}) - \left( \eta_{\mathbf{z}} - \frac{3\eta_{\mathbf{z}}^2 L_{\mathbf{z}}}{2} \right) \mathbb{E} \|\nabla_{\mathbf{z}} \mathcal{L}_\mu(\boldsymbol{\alpha}^{t+1}, \mathbf{z}^t)\|^2$$

$$+ \frac{3\eta_{\mathbf{z}}^2 L_{\mathbf{z}}}{2} \mathbb{E} \left( \frac{(4 + 2B)(d+4)}{B} \|\nabla_{\mathbf{z}} \mathcal{L}(\boldsymbol{\alpha}^{t+1}, \mathbf{z}^t)\|^2 + \frac{4(d+4)\sigma_{\mathbf{z}}^2}{B} + \frac{\mu^2 L_{\mathbf{z}}^2 (d+6)^3}{2} (\frac{1}{B} + 1) \right) + \mu^2 L_{\mathbf{z}} d$$

$$\leq - (\eta_{\boldsymbol{\alpha}} - \eta_{\boldsymbol{\alpha}}^2 L_{\boldsymbol{\alpha}}) \mathbb{E} \|\nabla_{\boldsymbol{\alpha}} \mathcal{L}(\boldsymbol{\alpha}^t, \mathbf{z}^t)\|^2 + \eta_{\boldsymbol{\alpha}}^2 L_{\boldsymbol{\alpha}} n (\frac{2\sigma_g^2}{I^2} + \frac{2\sigma_{\boldsymbol{\alpha}}^2}{B}) - \left( \eta_{\mathbf{z}} - \frac{3\eta_{\mathbf{z}}^2 L_{\mathbf{z}}}{2} \right) \mathbb{E} \|\nabla_{\mathbf{z}} \mathcal{L}_\mu(\boldsymbol{\alpha}^{t+1}, \mathbf{z}^t)\|^2$$

$$+ \frac{3\eta_{\mathbf{z}}^2 L_{\mathbf{z}}}{2} \mathbb{E} \left( 6(d+4) \|\nabla_{\mathbf{z}} \mathcal{L}(\boldsymbol{\alpha}^{t+1}, \mathbf{z}^t)\|^2 + \frac{4(d+4)\sigma_{\mathbf{z}}^2}{B} + 2\mu^2 L_{\mathbf{z}}^2 d^3 \right) + \mu^2 L_{\mathbf{z}} d$$

$$\leq - (\eta_{\boldsymbol{\alpha}} - \eta_{\boldsymbol{\alpha}}^2 L_{\boldsymbol{\alpha}}) \mathbb{E} \|\nabla_{\boldsymbol{\alpha}} \mathcal{L}(\boldsymbol{\alpha}^t, \mathbf{z}^t)\|^2 + \eta_{\boldsymbol{\alpha}}^2 L_{\boldsymbol{\alpha}} n (\frac{2\sigma_g^2}{I^2} + \frac{2\sigma_{\boldsymbol{\alpha}}^2}{B})$$

$$- \left( \eta_{\mathbf{z}} - \frac{3\eta_{\mathbf{z}}^2 L_{\mathbf{z}}}{2} \right) \mathbb{E} \left( \frac{1}{2} \mathbb{E} \|\nabla_{\mathbf{z}} \mathcal{L}(\boldsymbol{\alpha}^{t+1}, \mathbf{z}^t)\| - \|\nabla_{\mathbf{z}} \mathcal{L}(\boldsymbol{\alpha}^{t+1}, \mathbf{z}^t) - \nabla_{\mathbf{z}} \mathcal{L}_\mu(\boldsymbol{\alpha}^{t+1}, \mathbf{z}^t)\|^2 \right)$$

$$+ \frac{3\eta_{\mathbf{z}}^2 L_{\mathbf{z}}}{2} \mathbb{E} \left( 6(d+4) \|\nabla_{\mathbf{z}} \mathcal{L}(\boldsymbol{\alpha}^{t+1}, \mathbf{z}^t)\|^2 + \frac{4(d+4)\sigma_{\mathbf{z}}^2}{B} + 2\mu^2 L_{\mathbf{z}}^2 d^3 \right) + \mu^2 L_{\mathbf{z}} d$$

$$\leq - (\eta_{\boldsymbol{\alpha}} - \eta_{\boldsymbol{\alpha}}^2 L_{\boldsymbol{\alpha}}) \, \mathbb{E}\|\nabla_{\boldsymbol{\alpha}}\mathcal{L}(\boldsymbol{\alpha}^t, \mathbf{z}^t)\|^2 + \eta_{\boldsymbol{\alpha}}^2 L_{\boldsymbol{\alpha}} n(\frac{2\sigma_g^2}{I^2} + \frac{2\sigma_{\boldsymbol{\alpha}}^2}{B})$$

$$- \left( \eta_{\mathbf{z}} - \frac{3\eta_{\mathbf{z}}^2 L_{\mathbf{z}}}{2} \right) \mathbb{E} \left( \frac{1}{2}\mathbb{E}\|\nabla_{\mathbf{z}}\mathcal{L}(\boldsymbol{\alpha}^{t+1}, \mathbf{z}^t)\| - \frac{\mu^2}{4}L_{\mathbf{z}}^2(d+3)^3 \right)$$

$$+ \frac{3\eta_{\mathbf{z}}^2 L_{\mathbf{z}}}{2}\mathbb{E} \left( 6(d+4)\|\nabla_{\mathbf{z}}\mathcal{L}(\boldsymbol{\alpha}^{t+1}, \mathbf{z}^t)\|^2 + \frac{4(d+4)\sigma_{\mathbf{z}}^2}{B} + 2\mu^2 L_{\mathbf{z}}^2 d^3 \right) + \mu^2 L_{\mathbf{z}}d$$

$$= - (\eta_{\boldsymbol{\alpha}} - \eta_{\boldsymbol{\alpha}}^2 L_{\boldsymbol{\alpha}}) \, \mathbb{E}\|\nabla_{\boldsymbol{\alpha}}\mathcal{L}(\boldsymbol{\alpha}^t, \mathbf{z}^t)\|^2 + \eta_{\boldsymbol{\alpha}}^2 L_{\boldsymbol{\alpha}} n(\frac{2\sigma_g^2}{I^2} + \frac{2\sigma_{\boldsymbol{\alpha}}^2}{B}) - \left( \frac{\eta_{\mathbf{z}}}{2} - 10(d+4)\eta_{\mathbf{z}}^2 L_{\mathbf{z}} \right) \mathbb{E}\|\nabla_{\mathbf{z}}\mathcal{L}(\boldsymbol{\alpha}^{t+1}, \mathbf{z}^t)\|^2$$

$$+ \left( \eta_{\mathbf{z}} - \frac{3\eta_{\mathbf{z}}^2 L_{\mathbf{z}}}{2} \right) \frac{\mu^2}{4}L_{\mathbf{z}}^2(d+3)^3 + \frac{6(d+4)\eta_{\mathbf{z}}^2 L_{\mathbf{z}}\sigma_{\mathbf{z}}^2}{B} + 3\mu^2\eta_{\mathbf{z}}^2 L_{\mathbf{z}}^3 d^3 + \mu^2 L_{\mathbf{z}}d.$$

Inequality $(g)$ holds because $\mathbb{E}\|a+b\|^2 \leq 2\mathbb{E}\|a\|^2 + 2\mathbb{E}\|b\|^2$ and $\mathbb{E}\|a+b+c\|^2 \leq 3\mathbb{E}\|a\|^2 + 3\mathbb{E}\|b\|^2 + 3\mathbb{E}\|c\|^2$.

Inequality $(h)$ holds because each gradient with respect to $\boldsymbol{\alpha}_i$ would have a variance, then the lower bound of the variance need to multiply by $n$ because the token length is $n$, and they are independent.

Let $\eta_{\boldsymbol{\alpha}} = \dfrac{1}{2L_{\boldsymbol{\alpha}}}$ and $\eta_{\mathbf{z}} = \dfrac{1}{40(d+4)L_{\mathbf{z}}}$, we have

$$\mathbb{E}\mathcal{L}(\boldsymbol{\alpha}^{t+1}, \mathbf{z}^{t+1}) - \mathcal{L}(\boldsymbol{\alpha}^t, \mathbf{z}^t) \leq - \frac{\eta_{\boldsymbol{\alpha}}}{2}\mathbb{E}\|\nabla_{\boldsymbol{\alpha}}\mathcal{L}(\boldsymbol{\alpha}^t, \mathbf{z}^t)\|^2 - \frac{\eta_{\mathbf{z}}}{4}\mathbb{E}\|\nabla_{\mathbf{z}}\mathcal{L}(\boldsymbol{\alpha}^{t+1}, \mathbf{z}^t)\|^2 + \frac{n\sigma_g^2}{2L_{\boldsymbol{\alpha}}I^2}$$

$$+ \frac{n\sigma_{\boldsymbol{\alpha}}^2}{2L_{\boldsymbol{\alpha}}B} + L_{\mathbf{z}}d^2\mu^2 + \frac{\sigma_{\mathbf{z}}^2}{(d+4)L_{\mathbf{z}}B} + L_{\mathbf{z}}d\mu^2 + L_{\mathbf{z}}d\mu^2$$

$$= - \frac{1}{4L_{\boldsymbol{\alpha}}}\mathbb{E}\|\nabla_{\boldsymbol{\alpha}}\mathcal{L}(\boldsymbol{\alpha}^t, \mathbf{z}^t)\|^2 - \frac{1}{160(d+4)L_{\mathbf{z}}}\mathbb{E}\|\nabla_{\mathbf{z}}\mathcal{L}(\boldsymbol{\alpha}^{t+1}, \mathbf{z}^t)\|^2 + \frac{n\sigma_g^2}{2L_{\boldsymbol{\alpha}}I^2}$$

$$+ \frac{n\sigma_{\boldsymbol{\alpha}}^2}{2L_{\boldsymbol{\alpha}}B} + L_{\mathbf{z}}d^2\mu^2 + \frac{\sigma_{\mathbf{z}}^2}{(d+4)L_{\mathbf{z}}B} + 2L_{\mathbf{z}}d\mu^2.$$

Rearranging the formulation, we have

$$\frac{1}{4L_{\boldsymbol{\alpha}}}\mathbb{E}\|\nabla_{\boldsymbol{\alpha}}\mathcal{L}(\boldsymbol{\alpha}^t, \mathbf{z}^t)\|^2 + \frac{1}{160(d+4)L_{\mathbf{z}}}\mathbb{E}\|\nabla_{\mathbf{z}}\mathcal{L}(\boldsymbol{\alpha}^{t+1}, \mathbf{z}^t)\|^2$$

$$\leq \mathcal{L}(\boldsymbol{\alpha}^t, \mathbf{z}^t) - \mathbb{E}\mathcal{L}(\boldsymbol{\alpha}^{t+1}, \mathbf{z}^{t+1}) + \frac{n\sigma_g^2}{2L_{\boldsymbol{\alpha}}I^2} + \frac{n\sigma_{\boldsymbol{\alpha}}^2}{2L_{\boldsymbol{\alpha}}B} + L_{\mathbf{z}}d^2\mu^2 + \frac{\sigma_{\mathbf{z}}^2}{(d+4)L_{\mathbf{z}}B} + 2L_{\mathbf{z}}d\mu^2.$$

Let $L_{\max} = \max\{L_{\boldsymbol{\alpha}}, (d+4)L_{\mathbf{z}}\}$, $L_{\min} = \min\{L_{\boldsymbol{\alpha}}, (d+4)L_{\mathbf{z}}\}$, we have

$$\frac{1}{4L_{\max}}\mathbb{E}\|\nabla_{\boldsymbol{\alpha}}\mathcal{L}(\boldsymbol{\alpha}^t, \mathbf{z}^t)\|^2 + \frac{1}{160L_{\max}}\mathbb{E}\|\nabla_{\mathbf{z}}\mathcal{L}(\boldsymbol{\alpha}^{t+1}, \mathbf{z}^t)\|^2$$

$$\leq \mathcal{L}(\boldsymbol{\alpha}^t, \mathbf{z}^t) - \mathbb{E}\mathcal{L}(\boldsymbol{\alpha}^{t+1}, \mathbf{z}^{t+1}) + \frac{n\sigma_g^2}{2L_{\min}I^2} + \frac{n\sigma_{\boldsymbol{\alpha}}^2}{2L_{\min}B} + \frac{\sigma_{\mathbf{z}}^2}{L_{\min}B} + 3L_{\max}d\mu^2.$$

Let $\kappa = \dfrac{L_{\max}}{L_{\min}}$ and multiple sides with $160L_{\max}$, we have

$$\mathbb{E}\|\nabla_{\boldsymbol{\alpha}}\mathcal{L}(\boldsymbol{\alpha}^t, \mathbf{z}^t)\|^2 + \mathbb{E}\|\nabla_{\mathbf{z}}\mathcal{L}(\boldsymbol{\alpha}^{t+1}, \mathbf{z}^t)\|^2$$

$$\leq 160L_{\max}(\mathcal{L}(\boldsymbol{\alpha}^t, \mathbf{z}^t) - \mathbb{E}\mathcal{L}(\boldsymbol{\alpha}^{t+1}, \mathbf{z}^{t+1})) + \frac{80n\kappa\sigma_g^2}{I^2} + \frac{80n\kappa\sigma_{\boldsymbol{\alpha}}^2}{B} + \frac{160\kappa\sigma_{\mathbf{z}}^2}{B} + 480L_{\max}^2 d\mu^2.$$

Summing them up from $t=0$ to $T-1$, we have

$$\frac{1}{T}\sum_{t=0}^{T-1}\mathbb{E}\left(\|\nabla_{\boldsymbol{\alpha}}\mathcal{L}(\boldsymbol{\alpha}^t, \mathbf{z}^t)\|^2 + \|\nabla_{\mathbf{z}}\mathcal{L}(\boldsymbol{\alpha}^{t+1}, \mathbf{z}^t)\|^2\right)$$

$$\leq \frac{1}{T}\sum_{t=0}^{T-1}160L_{\max}(\mathcal{L}(\boldsymbol{\alpha}^t, \mathbf{z}^t) - \mathbb{E}\mathcal{L}(\boldsymbol{\alpha}^{t+1}, \mathbf{z}^{t+1})) + \frac{80n\kappa\sigma_g^2}{I^2} + \frac{80n\kappa\sigma_{\boldsymbol{\alpha}}^2}{B} + \frac{160\kappa\sigma_{\mathbf{z}}^2}{B} + 480L_{\max}^2 d\mu^2.$$

$$\leq \frac{160 L_{\max}(\mathcal{L}(\boldsymbol{\alpha}^0, \mathbf{z}^0) - \mathcal{L}^*)}{T} + \frac{80 n \kappa \sigma_g^2}{I^2} + \frac{80 n \kappa \sigma_{\boldsymbol{\alpha}}^2}{B} + \frac{160 \kappa \sigma_{\mathbf{z}}^2}{B} + 480 L_{\max}^2 d\mu^2.$$

Let $I = \mathcal{O}\left(\frac{\sqrt{n\kappa}\sigma_g}{\epsilon}\right), B = \mathcal{O}\left(\max\{\frac{n\kappa\sigma_{\boldsymbol{\alpha}}^2}{\epsilon^2}, \frac{\kappa\sigma_{\mathbf{z}}^2}{\epsilon^2}\}\right), \mu = \mathcal{O}\left(\frac{1}{L_{max}}\sqrt{\frac{\epsilon^2}{d}}\right)$ and $T = \mathcal{O}\left(\frac{L_{max}}{\epsilon^2}\right)$, we have

$$\mathbb{E}\left(\|\nabla_{\boldsymbol{\alpha}}\mathcal{L}(\boldsymbol{\alpha}^t, \mathbf{z}^t)\|^2 + \|\nabla_{\mathbf{z}}\mathcal{L}(\boldsymbol{\alpha}^{t+1}, \mathbf{z}^t)\|^2\right) \leq \mathcal{O}(\epsilon^2).$$

$\square$

# B  MORE EXPERIMENTAL DETAILS

## B.1  MANUAL TEMPLATES

Table 4: Input templates, and output label words used in RoBERTa-large. $\langle S \rangle$ represents the sentences in the dataset. [MASK] represents the mask token.

| Task | Dataset | Input Template | Output Label Words |
|------|---------|----------------|--------------------|
| acceptability | CoLA | $\langle S \rangle$ correct? [MASK]. | no, yes |
| NLI | MNLI | $\langle S_1 \rangle\langle S_2 \rangle$ entailment? [MASK]. | no, maybe, yes |
| NLI | QNLI | $\langle S_1 \rangle\langle S_2 \rangle$ entailment? [MASK]. | no, yes |
| NLI | SNLI | $\langle S_1 \rangle\langle S_2 \rangle$ entailment? [MASK]. | no, maybe, yes |
| NLI | WNLI | $\langle S_1 \rangle\langle S_2 \rangle$ entailment? [MASK]. | no, yes |

Table 5: Input templates, and output label words used in GPT2-XL and Llama3. $\langle S \rangle$ represents the sentences in the dataset.

| Task | Dataset | Input Template | Output Label Words |
|------|---------|----------------|--------------------|
| acceptability | CoLA | $\langle S \rangle$ correct? | no, yes |
| NLI | MNLI | $\langle S_1 \rangle\langle S_2 \rangle$ entailment? | no, maybe, yes |
| NLI | QNLI | $\langle S_1 \rangle\langle S_2 \rangle$ entailment? | no, yes |
| NLI | SNLI | $\langle S_1 \rangle\langle S_2 \rangle$ entailment? | no, maybe, yes |
| NLI | WNLI | $\langle S_1 \rangle\langle S_2 \rangle$ What is the relation? | no, yes |

## B.2  HYPERPARAMETERS

Table 6: Main hyperparameters used in our algorithms.

| Hyperparameter | RoBERTa-large | GPT2-XL | Llama3 |
|----------------|---------------|---------|--------|
| query limit | 4000 | 2000 | 1000 |
| train batch size $B$ | 32 | 16 | 8 |
| prompt length $n$ | $\{50, 20\}$ | $\{50, 20\}$ | $\{50, 20\}$ |
| small positive constant $\mu$ | | 0.01 | |
| sample times $I_1$ | | 20 | |
| sample times $I_2$ | | 10 | |
| temperature $\tau$ | | 1.0 | |
| vocabulary size $|\mathcal{V}|$ | | 100 | |
| intrinsic dimensionality $d$ | | 500 | |

## C  MORE ABLATION STUDIES

**Effect of Optimizing Initial Prompt.** To evaluate the impact of optimizing the initial discrete prompt, we conducted an ablation study where we compared the performance of ZO-PoG using optimized and randomly initialized initial prompts. Specifically, we investigated whether learning a task-specific distribution for selecting initial prompt tokens, instead of random selection, contributes to performance improvements across downstream tasks. To achieve this, we replaced sampling from Gumbel-Softmax with randomly selected tokens from the PTM vocabulary. Both setups employed the same continuous prompt optimization pipeline to isolate the impact of initial prompt optimization. The comparative results are reported in Table 7 and Table 8. The comparative results indicate that models initialized with optimized discrete prompts consistently outperformed those using random initialization across all datasets.

Table 7: Comparison of the average accuracy (%) between randomly sampled $\mathbf{p}_0$ and optimized the $\mathbf{p}_0$ (ZO-PoG) on RoBERTa-Large in 16-shot (per class) setting.

| Len | Method | CoLA | MNLI | QNLI | SNLI | WNLI |
|---|---|---|---|---|---|---|
| 20 | random $\mathbf{p}_0$ (seed = 14) | 52.64 | 41.65 | 54.57 | 39.06 | 47.89 |
| | random $\mathbf{p}_0$ (seed = 42) | 46.69 | 41.31 | 53.14 | 39.65 | 52.11 |
| | random $\mathbf{p}_0$ (seed = 81) | 55.23 | 42.54 | 51.18 | 38.32 | 42.25 |
| | ZO-PoG (mean) | **56.79** | **43.29** | **55.44** | **39.12** | **53.99** |

Table 8: Comparison of the average accuracy (%) between randomly sampled $\mathbf{p}_0$ and optimized the $\mathbf{p}_0$ (ZO-PoG) on Llama3 in 16-shot (per class) setting.

| Len | Method | CoLA | MNLI | QNLI | SNLI | WNLI |
|---|---|---|---|---|---|---|
| 20 | random $\mathbf{p}_0$ (seed = 14) | 50.43 | 34.55 | 53.21 | 37.41 | 56.34 |
| | random $\mathbf{p}_0$ (seed = 42) | 37.20 | 38.05 | 53.49 | 37.09 | 43.66 |
| | random $\mathbf{p}_0$ (seed = 81) | 43.53 | 35.39 | 53.41 | 36.16 | 59.15 |
| | ZO-PoG (first run) | 41.71 | 41.16 | 53.14 | 39.44 | 56.34 |
| | ZO-PoG (second run) | 49.28 | 39.08 | 55.67 | 35.01 | 57.75 |
| | ZO-PoG (third run) | 53.40 | 36.26 | 58.91 | 40.33 | 63.38 |
| | random $\mathbf{p}_0$ (max) | 50.43 | 38.05 | 53.49 | 37.41 | 59.15 |
| | ZO-PoG (max) | 53.40 | 41.16 | 58.91 | 40.33 | 63.38 |
| | random $\mathbf{p}_0$ (mean) | 43.72 | 36.00 | 53.37 | 36.89 | 53.05 |
| | ZO-PoG (mean) | **48.13** | **38.84** | **55.90** | **38.26** | **59.15** |

In order to further verify the sensitivity of prompt initialization, we also tried the following different sampling strategies for $\mathbf{p}_0$: 1) we sample the same token for each position of the fixed length prompt. 2) we uniformly randomly sample from the vocabulary (the same way as that sampled with random seeds). 3) We sample the continuous tokens of fixed length from the PTM vocabulary (this is the same way as implemented in BBT). We have also tried to directly sample $\mathbf{p}_0$ in the embedding space as initialization through 4) Gaussian distribution (with the same mean and variance value for the sampling of random matrix $\mathbf{A}$) and 5) uniform distribution within a fixed region (mean standard deviation of the sampling of random matrix $\mathbf{A}$). 6) Orthogonal initialization: each token embedding in are orthogonal with each other. We report the accuracy with three different random seeds in Table 9.

**Effect of Prompt Length.** To investigate how the prompt length affects the performance of our proposed ZO-PoG framework, we experimented with prompt lengths of 10, 20, 30, 40, 50 tokens on the performance of Llama for QNLI dataset. The results in Table 10 show that the relationship between prompt length and model performance is non-linear. Short prompts may demonstrate limited representational capacity, leading to suboptimal performance and longer prompts do not necessarily produce better result. The observed trends suggest that a moderately long prompt provides a good trade-off between capacity and learnability.

**Effect of Optimization Strategy.** To verify the necessity and effectiveness of alternating optimization, we conducted an ablation study to compare three optimization strategies: 1) Alternating

Table 9: Accuracy (%) on RoBERTa-Large in 16-shot (per class) setting with different initialization strategy.

| | | CoLA | | | MNLI | | | QNLI | | |
|---|---|---|---|---|---|---|---|---|---|---|
| Initialization Strategy | Seed | 14 | 42 | 81 | 14 | 42 | 81 | 14 | 42 | 81 |
| prompt with same tokens | | 46.40 | 44.87 | 45.93 | 41.37 | 42.64 | 41.40 | 55.30 | 52.17 | 52.70 |
| Uniform sampling (Vocabulary) | | 52.64 | 46.69 | 55.23 | 41.65 | 41.31 | 42.54 | 54.57 | 53.14 | 51.18 |
| continuous token in Vocabulary | | 45.54 | 49.19 | 42.57 | 43.29 | 36.18 | 43.68 | 51.38 | 54.37 | 52.00 |
| Gaussian sampling (embedding) | | 50.24 | 49.28 | 53.4 | 38.31 | 38.34 | 37.54 | 52.04 | 52.44 | 53.96 |
| Uniform sampling (embedding) | | 48.03 | 43.53 | 54.75 | 38.67 | 42.22 | 35.59 | 52.39 | 50.52 | 51.33 |
| Orthogonal initialization | | 42.57 | 37.87 | 53.98 | 35.84 | 40.89 | 39.68 | 53.54 | 52.02 | 52.83 |

| | | SNLI | | | WNLI | | | | | |
|---|---|---|---|---|---|---|---|---|---|---|
| Initialization Strategy | Seed | 14 | 42 | 81 | 14 | 42 | 81 | | | |
| prompt with same tokens | | 36.99 | 40.06 | 38.59 | 42.25 | 42.25 | 38.03 | | | |
| Uniform sampling (Vocabulary) | | 39.06 | 39.65 | 38.32 | 47.89 | 52.11 | 42.25 | | | |
| continuous token in vocabulary | | 38.91 | 35.15 | 34.33 | 42.25 | 56.33 | 53.52 | | | |
| Gaussian sampling (embedding) | | 35.38 | 35.87 | 36.23 | 45.07 | 50.7 | 42.25 | | | |
| Uniform sampling (embedding) | | 38.84 | 34.22 | 33.58 | 49.3 | 52.11 | 54.93 | | | |
| Orthogonal initialization | | 37.38 | 34.00 | 35.01 | 54.93 | 45.07 | 54.93 | | | |

Table 10: Ablation study of prompt lengths on Llama3 in 16-shot (per class) setting with the QNLI dataset.

| Dataset | Prompt length | 10 | 20 | 30 | 40 | 50 | 80 |
|---|---|---|---|---|---|---|---|
| QNLI | | 53.17 | 55.90 | 56.13 | 55.83 | 55.60 | 54.73 |

optimization: The proposed strategy alternates between discrete and continuous prompt optimization across multiple iterations. 2) Single-round Optimization: This approach involves performing one round of soft-prompt optimization followed by one round of zeroth-order optimization without alternating iterations. 3) Joint optimization: Both soft prompt and discrete probability distribution are optimized simultaneously using zeroth-order optimization. The results in Table 11 and 12 verify the design of the alternating optimization strategy. By leveraging the strengths of both policy gradient and zeroth-order optimization iteratively, ZO-PoG achieves superior performance

Table 11: The comparison of the average accuracy between one round ZO + one round Policy gradient and ZO-PoG on Llama3 on 16-shot (per class). The best results are underlined. **Len** represents the prompt length.

| Len | Method | CoLA | MNLI | QNLI | SNLI | WNLI |
|---|---|---|---|---|---|---|
| 20 | one round + one round | 43.50 | 37.48 | 50.58 | 34.29 | 54.93 |
| | ZO-PoG (ours) | **48.13** | **38.84** | **55.90** | **38.26** | **59.15** |
| 50 | one round + one round | 42.25 | 36.13 | 55.09 | 33.70 | 54.93 |
| | ZO-PoG (ours) | **46.50** | **37.19** | **55.60** | **38.12** | **58.22** |

# D EXPERIMENTAL RESULTS ON BLACK-BOX LARGE LANGUAGE MODELS

## D.1 ADAPT ZO-POG TO BLACK-BOX LARGE LANGUAGE MODELS

Modern black-box large language models (LLMs), such as GPT-4, offer powerful capabilities but restrict user access to their internal components, such as the embedding space. These models accept only discrete input tokens, which limits direct optimization in the continuous embedding space. Additionally, feedback from black-box APIs is available only through forward evaluations, further complicating optimization.

Table 12: The comparison of the average accuracy between joint optimization with zeroth-order optimization and ZO-PoG on RoBERTa-Large in 16-shot setting. The best results are underlined. **Len** represents the prompt length.

| Len | Method | CoLA | MNLI | QNLI | SNLI | WNLI |
|-----|--------|------|------|------|------|------|
| 20 | joint optimization | 52.33 | 41.00 | 54.19 | 35.47 | 46.95 |
| | ZO-PoG (ours) | **56.79** | **43.29** | **55.44** | **39.12** | **53.99** |
| 50 | joint optimization | 53.05 | 41.92 | **55.56** | 35.74 | 46.00 |
| | ZO-PoG (ours) | **56.18** | **42.53** | 55.33 | **41.99** | **56.34** |

---

**Algorithm 2** ZO-PoG with black-box Large Language Models

---

**Input:** Parameters of categorical distributions $\boldsymbol{\alpha}_1, \cdots, \boldsymbol{\alpha}_n$, temperature parameter $\tau$, black-box large language models $f$, surrogate embedding model $\mathbf{e}(\cdot)$, downstream dataset $\mathcal{D}$, mini-batch size $B$, learning rates $\eta_{\boldsymbol{\alpha}}, \eta_{\mathbf{z}}$ and sample times $I_1, I_2$.

1: Initialize $\boldsymbol{\alpha}_1^0, \cdots, \boldsymbol{\alpha}_n^0, \mathbf{z}^0$
2: **for** $t$ in 0 to $T-1$ **do**
3:   Draw a mini-batch $\mathcal{S}_t = \{X_t, Y_t\} = \{x_i, y_i\}_{i=1}^B$ from $\mathcal{D}$
4:   **for** $k$ in 1 to $I_1$ **do**
5:    Sample $j_{1,t}^{(k)} \sim \text{GS}(\boldsymbol{\alpha}_1^t, \tau), \cdots, j_{n,t}^{(k)} \sim \text{GS}(\boldsymbol{\alpha}_n^t, \tau)$
6:    $\mathbf{p}_{0,t}^{(k)} = \mathbf{e}(t_1^{(k)}, \cdots, t_n^{(k)}) = \mathbf{e}(\mathcal{V}[j_1^{(k)}], \cdots, \mathcal{V}[j_n^{(k)}])$
7:   $\mathcal{L}_{\text{avg}} = \frac{1}{I_1} \sum_{k=1}^{I_1} \mathcal{L}(f(\mathbf{e}^{-1}(\mathbf{A}\mathbf{z}^t + \mathbf{p}_{0,t}^{(k)}); X_t), Y_t)$
8:   **for** $i$ in 1 to $n$ **do**
9:    $g_{\boldsymbol{\alpha}_i^t}^{vr} = \frac{1}{I_1-1} \sum_{k=1}^{I_1} (\mathcal{L}(f(\mathbf{e}^{-1}(\mathbf{A}\mathbf{z}^t + \mathbf{p}_{0,t}^{(k)}); X_t), Y_t) - \mathcal{L}_{\text{avg}}) \nabla_{\boldsymbol{\alpha}_i} \log P(t_i^{(k)}|\boldsymbol{\alpha}_i^t)$
10:    $\boldsymbol{\alpha}_i^{t+1} \leftarrow \boldsymbol{\alpha}_i^t - \eta_{\boldsymbol{\alpha}} g_{\boldsymbol{\alpha}_i^t}^{vr}$        ▷ Update discrete vocabulary distribution
11:   **for** $l$ in 1 to $I_2$ **do**
12:    Sample $j_{1,t}^{(l)} \sim \text{GS}(\boldsymbol{\alpha}_1^{t+1}, \tau), \cdots, j_{n,t}^{(l)} \sim \text{GS}(\boldsymbol{\alpha}_n^{t+1}, \tau)$
13:    $\mathbf{p}_{0,t}^{(l)} = \mathbf{e}(t_{1,t}^{(l)}, \cdots, t_{n,t}^{(l)}) = \mathbf{e}(\mathcal{V}[j_{1,t}^{(l)}], \cdots, \mathcal{V}[j_{n,t}^{(l)}])$
14:    Sample $\mathbf{u}_t^{(l)} \sim \mathcal{N}(0, \mathbf{I}_d)$
15:    $\mathcal{L}_+^{(l)} = \mathcal{L}(f(\mathbf{e}^{-1}(\mathbf{A}(\mathbf{z}^t + \mu\mathbf{u}_t^{(l)}) + \mathbf{p}_{0,t}^{(l)}); X_t), Y_t)$
16:    $\mathcal{L}_-^{(l)} = \mathcal{L}(f(\mathbf{e}^{-1}(\mathbf{A}(\mathbf{z}^t - \mu\mathbf{u}_t^{(l)}) + \mathbf{p}_{0,t}^{(l)}); X_t), Y_t)$
17:    $g_{\mathbf{z}^t}^{(l)} = (\mathcal{L}_+^{(l)} - \mathcal{L}_-^{(l)})/(2\mu) \cdot \mathbf{u}_t^{(l)}$
18:   $g_{\mathbf{z}^t} = \frac{1}{I_2} \sum_{l=1}^{I_2} g_{\mathbf{z}^t}^{(l)}$
19:   $\mathbf{z}^{t+1} = \mathbf{z}^t - \eta_{\mathbf{z}} g_{\mathbf{z}^t}$         ▷ Update continuous prompt
**Return:** $\boldsymbol{\alpha}_1^T, \cdots, \boldsymbol{\alpha}_n^T, \mathbf{z}^T$

---

To address these challenges, we propose an enhanced version of ZO-PoG tailored for discrete-token-only LLMs. The new algorithm leverages a surrogate embedding model $\mathbf{e}(\cdot) : \mathcal{V}^n \to \mathbb{R}^D$ to approximate the black-box LLM's embedding space and enables efficient optimization using a projection function that maps continuous embeddings back to discrete tokens. We could adopt a pre-trained open-sourced as the surrogate embedding model (e.g., RoBERTa, GPT-2). This approach ensures the method can adapt to real-world LLM scenarios without compromising performance.

The proposed algorithm consists of a preprocessing step for surrogate embedding computation and an alternating optimization framework that integrates both discrete and continuous prompt optimization. Let $\mathcal{V}_s$ denote the full vocabulary of the surrogate embedding model. Before the training of the algorithm, we will first need to compute the embedding $\{\mathbf{e}(v)|v \in \mathcal{V}_s\}$ for all tokens in $\mathcal{V}_s$. Then we could construct a lookup table $\mathcal{E}_s = \{(v, \mathbf{e}(v)) \mid v \in \mathcal{V}_s\}$ for efficient access during training. During training, we define a projection function $\mathbf{e}^{-1}(\cdot) : \mathbb{R}^D \to \mathcal{V}_s^n$ that map a learned embedding $\mathbf{p} \in \mathbb{R}^D$ back to the nearest or the most similar token in $\mathcal{V}_s$. In this work, we choose the cosine

similarity as the projection function, i.e.,

$$\mathbf{e}^{-1}(\mathbf{p}) = \arg \max_{v \in \mathcal{V}_s^n} \frac{\langle \mathbf{p}, \mathbf{e}(v) \rangle}{\|\mathbf{p}\| \, \|\mathbf{e}(v)\|}, \tag{22}$$

which is a token-wise projection operation.

## D.2 EXPERIMENTS ON MATH TASK

To provide a detailed evaluation of ZO-PoG's effectiveness on tasks requiring advanced reasoning, we conducted experiments on a challenging mathematical problem-solving dataset GSM8K (Cobbe et al., 2021) in the 64-shot setting. In addition to comparisons with prompt learning baselines, we have also included comparisons with instruction optimization methods InstructZero (Chen et al., 2024) and TRIPLE (Shi et al., 2024). For prompt learning methods, the prompt prefix is optimized, which is concatenated with a manual prompt with Chain-of-Thought (CoT), i.e., Let's think step by step. We choose open-sourced GPT-2 as our surrogate embedding model. For instruction optimization baselines, the entire instruction is optimized. The input templates are shown in Table 13. In order to enhance exploration, we also adopt the single-point gradient estimation for ZO-PoG in this task, i.e., replacing Line 17 of Algorithm 2 with $g_{\mathbf{z}^t}^{(l)} = \mathcal{L}_+^{(l)}/\mu \cdot \mathbf{u}_t^{(l)}$. The experimental results on the GPT-4 are presented in Table 14. Our method achieves the most significant performance improvement compared to Manual Prompt with the Chain of Thought in the context of prompt learning baselines and also demonstrates superior performance over instruction optimization methods.

Table 13: Input templates used in GSM8K dataset. $\langle Q \rangle$ represents the questions in the dataset. $\backslash n$ represents the line break.

| Method | Input Template |
|---|---|
| Prompt Learning | `[Prompt]` Let's think step by step.$\backslash n$ Question: $\langle Q \rangle$ $\backslash n$ Answer: |
| Instruction Optimization | `[Instruction]`$\backslash n$ Question: $\langle Q \rangle$ $\backslash n$ Answer: |

Table 14: Averaged scores of baselines and Zo-PoG on the GSM8K dataset using GPT-4.

| Method / Dataset | MP CoT | BBT | BDPL | TRIPLE | InstructZero | ZO-PoG (ours) |
|---|---|---|---|---|---|---|
| GSM8K | 86.48 | 78.22 | 85.77 | 70.26 | 89.16 | 89.94 |

