# OpenReview forum: "Collaborative Discrete-Continuous Black-Box Prompt Learning for Language Models"
_ICLR.cc/2025/Conference — ICLR 2025 Poster_

### Official Review · Reviewer_hHWU · 2024-11-02

**Soundness:** 2
**Presentation:** 3
**Contribution:** 2
**Rating:** 5
**Confidence:** 4

**Summary:**

This paper studies the problem of soft prompt optimization for pre-trained language models. In some previous works, random initialization is applied to a soft prompt which will be added to a low-dimensional vector's random projection. The authors proposed to optimize this random initialization as they believe this may lead to suboptimal performance. Technically, an alternating optimization method is proposed where the random initialization is optimized via a policy gradient method over token distributions, and the low-dimensional vector $z$ is optimized through zeroth-order optimization. Detailed convergence analysis is provided and empirical results are shown to justify the effectiveness of the proposed method.

**Strengths:**

- This work improves previous literature with special consideration on optimizing the initialization of soft prompts. The mixing of discrete prompt and soft prompt optimization is also innovative.
- Theoretical analysis of convergence is given to justify the design and principle of the proposed alternating optimization method.

**Weaknesses:**

- The authors stated their focus is on PTMs that can be interacted solely via APIs. However, most commercial models nowadays do not open embedding access,  which makes the soft prompt tuning method not practical and thus less significant. Especially in the experimental section, only a few white-box models are included and no black-box models are not considered.
- The recent literature on prompt optimization is not included. For example, InstructZero [1], ZOPO [2], and TRIPLE [3] were proposed to use a derivative-free method (Bayesian optimization or zeroth-order method) to optimize soft or discrete prompts. ZOPO and TRIPLE also project the discrete prompts into an embedding space to conduct optimization. Those methods need to be at least discussed and possibly compared.

[1] Chen, L., Chen, J., Goldstein, T., Huang, H., & Zhou, T. (2023). Instructzero: Efficient instruction optimization for black-box large language models.

[2] Hu, W., Shu, Y., Yu, Z., Wu, Z., Lin, X., Dai, Z., ... & Low, B. K. H. (2024). Localized zeroth-order prompt optimization.

[3] Shi, C., Yang, K., Yang, J., & Shen, C. (2024). Best arm identification for prompt learning under a limited budget.


- A strong motivation for this work is to improve the random initialization which is claimed to lead to a suboptimal performance. However, no empirical or theoretical justification is given. I suggest the authors conduct some ablation studies to demonstrate the sensitivity of prompt optimization performances given random initialization (number of random initialization or different random seeds).
- Why alternating optimization is necessary? Could the author demonstrate the difference between <just using one round of soft-prompt optimization + one round of zeroth-order optimization> and <the alternating optimization strategy>? I also feel a joint optimization is feasible here, where the low-dimensional $z$ and the encoded probability of the initialization prompt can be jointly optimized by zeroth-order optimization. These points should be helpful for justifying why the alternating optimization is designed.
- Assumption 1 normally does not hold in practice, especially when the loss function only produces discrete values for some NLP tasks, e.g., accuracy. I understand this assumption has to be made for the theory to work, but at least some level of justification should be given.
- Limited empirical results.
    - More representative baselines proposed in recent years should be considered.
    - Only one benchmark (i.e., GLUE) is considered. I expect to see comparisons on other more practical benchmarks, such as mathematical reasoning task GSM8K and some text generation tasks.

**Questions:**

See the questions above.

---

> ### Author Response · Authors · 2024-11-22
> **Response to Reviwer hHWU [Part 1]**
>
> We appreciate the time and effort put into reviewing our manuscript. Please find our responses to your concerns below:
>
> **weakness 1:** Thank you for pointing this problem out. In the revised manuscript, we have adapted our method to the real black-box large language models. To tackle the problem having no access to the embedding space of black-box LLMs, we introduce to use an open-sourced embedding model to construct the mapping between the whole vocabulary and their embedding. To be specific, let $V_{s}$ denote the full vocabulary of the surrogate embedding model. Before the training of the algorithm, we will first need to compute the embedding ${ e(v) | v \in V_s }$ for all tokens in $V_{s}$. Then we could construct a lookup table $\{(v, e(v)) \mid v \in V_s\}$ for efficient access during training. During training, we define a projection function $e^{-1}(\cdot): R^D \to V_s^n$ that map a learned embedding $p \in R^D$ back to the nearest or the most similar token in $V_s$. In this work, we choose the cosine similarity as the projection function, i.e.,
> $$
>     e^{-1} (p) =  argmax_{v \in V_s^n} \frac{\< p, e(v)\>}{|| p ||  || e(v)||},
> $$
> which is a token-wise projection operation. By projecting the soft prompt back to the token space, we reformulated our approach to operate purely within the constraints of API-based interactions. To demonstrate the practicality and effectiveness of the updated method, we conducted additional experiments using GPT-4. The results, included in Appendix D.2 of the revised manuscript, show that our method achieves competitive performance on the mathematical reasoning task GSM8K while requiring no access to embeddings or model internals. This extension strengthens the significance of our work by aligning it with the limitations of modern commercial LLM APIs.
>
> **weakness 2:** Thank you for pointing out the recent advancements in black-box prompt optimization, including InstructZero, ZOPO, and TRIPLE. We agree that these works are highly relevant and should be discussed to provide a more comprehensive context for our approach. We have revised the manuscript to include these methods in the related work section. For comparison, we conducted experiments on mathematical reasoning task GSM8K and compared our method with two black-box instruction optimization baselines: TRIPLE and InstructZero. The new results in Appendix D.2 demonstrate superior performance over these two instruction optimization methods. Note that, since our method are not dedicatedly designed for instruction optimization, the optimized discrete prompt may not be semantically meaningful. This is because our method are optimized directly for the downstream performance and didn't consider the semantic constraint, which may be a limitation of our method. We will consider this as a future work. However, considering the efficiency and scalability of our black-box prompt optimization method, it is still worthy of optimizing a task-specific prompt.
>
> **weakness 3:** Thank you this question, to evaluate the impact of optimizing the initial discrete prompt, we conducted an ablation study where we compared the performance of ZO-PoG using optimized and randomly initialized initial prompts. Specifically, we investigated whether learning a task-specific distribution for selecting initial prompt tokens, instead of random selection, contributes to performance improvements across downstream tasks. To achieve this, we replaced sampling from Gumbel-Softmax with randomly selected tokens from the PTM vocabulary. Both setups employed the same continuous prompt optimization pipeline to isolate the impact of initial prompt optimization.
> The comparative results are reported in the following table. The comparative results indicate that models initialized with optimized discrete prompts consistently outperformed those using random initialization across all datasets.
>
> ### Table: Comparison of the average accuracy (%) between randomly sampled $p_0$ and optimized $p_0$ (Zo-PoG) on Llama3 in 16-shot (per class)
>
> | Len | Method               | CoLA   | MNLI   | QNLI   | SNLI   | WNLI   |
> |-----|----------------------|--------|--------|--------|--------|--------|
> | 20  | w/o optimize $p_0$   | 43.52  | 32.71  | 53.37  | 36.89  | 53.05  |
> |     | **Zo-PoG (ours)**    | **48.13** | **35.33** | **55.90** | **38.26** | **59.15** |

---

> ### Author Response · Authors · 2024-11-22
> **Response to Reviwer hHWU [Part 2]**
>
> **weakness 4:** "why alternating optimization?"
>
> Thank you for your valuable feedback. Alternating optimization was designed to leverage the complementary strengths of soft-prompt and continuous prompt optimization while mitigating their individual limitations. Soft-prompt optimization aligns the initialization with task-specific distributions, while continuous optimization fine-tunes embeddings for finer-grained task adjustments. By alternating, each stage benefits from the improvements made by the other, avoiding suboptimal convergence seen in single-round optimization.
>
> Thank you for pointing out the potential of jointly optimizing the Gumbel-Softmax parameters and soft prompts using zeroth-order optimization. In our current approach, we use a policy gradient-based method, as it is specifically designed to handle discrete probability distributions. While zeroth-order optimization for Gumbel-Softmax parameters is theoretically feasible, it may introduce challenges such as higher gradient variance.
>
> To further justify our design, we have included an experimental comparison of single-round optimization, joint optimization and alternating optimization in the revised manuscript.
>
> ### Table: Comparison of average accuracy between one round ZO + one round Policy Gradient and ZO-PoG on Llama3 in 16-shot (per class)
> *The best results are underlined. **Len** represents the prompt length.*
>
> | Len | Method                    | CoLA   | MNLI   | QNLI   | SNLI   | WNLI   |
> |-----|---------------------------|--------|--------|--------|--------|--------|
> | 20  | one round $+$ one round   | 43.50  | 34.93  | 50.58  | 34.29  | 54.93  |
> |     | **ZO-PoG (ours)**         | **48.13** | **35.33** | **55.90** | **38.26** | **59.15** |
> | 50  | one round $+$ one round   | 42.25  | 35.18  | 55.09  | 33.70  | 54.93  |
> |     | **ZO-PoG (ours)**         | **46.50** | **35.30** | **55.60** | **38.12** | **58.22** |
>
> ### Table: Comparison of average accuracy between joint optimization with zeroth-order optimization and ZO-PoG on RoBERTa-Large in 16-shot setting
> *The best results are underlined. **Len** represents the prompt length.*
>
> | Len | Method                | CoLA   | MNLI   | QNLI   | SNLI   | WNLI   |
> |-----|-----------------------|--------|--------|--------|--------|--------|
> | 20  | joint optimization    | 52.33  | 29.79  | 54.19  | 35.47  | 46.95  |
> |     | **ZO-PoG (ours)**     | **56.79** | **35.11** | **55.44** | **39.12** | **53.99** |
> | 50  | joint optimization    | 53.05  | 28.07  | **55.56** | 35.74  | 46.00  |
> |     | **ZO-PoG (ours)**     | **56.18** | **35.46** | 55.33  | **41.99** | **56.34** |
>
> **weakness 5:**
>
> Thank you for your observation regarding Assumption 1 and its limitations in scenarios where the loss function is discrete, such as accuracy. In this paper, we focus on cross-entropy loss, which is a continuous and differentiable function commonly used as a surrogate for discrete metrics in NLP tasks. The smoothness assumption is valid for cross-entropy loss due to its Lipschitz continuous gradients and is essential for the theoretical convergence guarantees we provide. In the literature of zeroth-order fine-tuning of LLMs, such assumption was also utilized for the theoretical analysis [1].
> Moreover, the purpose of this part is not solely to establish rigorous theoretical results but also to gain insights into the behavior of our method. We believe that selectively incorporating these assumptions can yield valuable guidance for developing more principled and effective approaches to black-box optimization, which could also helpful to optimize non-differentiable objectives like accuracy and F1 score.
>
>
> [1] Fine-Tuning Language Models with Just Forward Passes. NeurIPS 2023.
>
> **weakness 6:**
> Thanks for your suggestion. We have added the GSM8K dataset to show our algorithm performance on the math problems. In this experiment, we used GPT-4 to adapt our algorithm to a true black-box model.
>
> ### Table: Averaged scores of baselines and Zo-PoG on the GSM8K dataset using GPT-4
>
> | Dataset | MP CoT | BBT   | BDPL  | TRIPLE | InstructZero | Zo-PoG (ours) |
> |---------|--------|-------|-------|--------|--------------|---------------|
> | GSM8K   | 86.48  | 78.22 | 85.77 | 70.26  | 89.16        | 89.94         |
>
> We sincerely appreciate the reviewer’s insightful feedback. If there are any additional concerns or questions, please feel free to let us know.

---

> ### Author Response · Authors · 2024-11-25
>
> Dear reviewer, thank you again for your time and effort put into reviewering our manuscript. Please let us know if our responses have addressed your concerns. If you have remaining concerns, please don't hestitate to let us know and we are happy to address them. Thank you!

---

> ### Author Response · Authors · 2024-11-27
>
> Dear Reviewer hHWU
>
> Thank you again for your time and effort dedicated to reviewing our paper. Please let us know if our responses have addressed your concerns. If you have remaining concerns, please don't hestitate to let us know and we are happy to address them.
>
> With best wishes,
>
> Authors

---

> ### Author Response · Authors · 2024-11-29
>
> Dear Reviewer hHWU,
>
> Thank you again for your time and effort dedicated to reviewing our paper. As the discussion phase is nearing its end, please let us know if our responses have addressed your concerns. If you have remaining concerns or questions, please don't hestitate to let us know and we are happy to discuss with you.
>
> Best regards,
>
> The Authors

---

> > ### Comment · Reviewer_hHWU · 2024-12-01
> >
> > Dear Authors,
> >
> > Thank the authors for providing the additional results and modifications which I think could strengthen the paper if the results and presentations are carefully consolidated. For Weakness 1: Although the authors proposed a fix for their paper for tacking black-box LLMs, the novelty still remains an issue, where the idea of steering a white-box model’s embedding is also very similar to InstructZero [1], and constructing an embedding lookup table is very similar to ZOPO [2].
> >
> > For Weakness 3: I suggest the authors run an ablation with different random initializations to see what the varying performances could be. This would be a direct motivation for optimizing the initialization.
> >
> > For Weakness 4: it does not make sense to me why the authors compare ZO-PoG with one round + one round, where the optimization budgets are not fair as ZO-PoG uses more total rounds of optimizations. Does the author mean to equally split the total optimization budget to ZO and PG? It is surprising that joint optimization actually performs much worse in experiments, I suspect it may be the high dimensional issue or the high gradient variance that the authors mentioned. The interpretation of this result should be discussed more in line 1115-1123.

---

> ### Author Response · Authors · 2024-12-01
> **Responses to the follow-up questions [Part 1]**
>
> Dear reviewer,
>
> Thank you so much for acknowledging our additional results and your remaining questions.
>
> First, we want to emphasize that the main focus and novelty of this work is to solve the problem of performance gap of black-box prompt learning induced by the random initialization of $p_0$. In order to tackle this problem, we propose to collaboratively optimize the discrete prompt in the token space and the continuous soft prompt in the embedding space. In order to adapt our method to the setting of the black-box large language models, it should be a necessity and also a natural idea to project the optimized soft prompt back to the token space by using the white-box large language models. Moreover, our projection method is a bit different from the one used in ZOPO, which compares the Euclidean distance of the optimized soft prompt and the candidate prompt embedding. As a comparison, we implement the projection by computing the token-wise cosine similarity of token embedding and the whole vocabulary embedding.
>
> For Weakness 3, thank you for your clarification. In our original response to this question, we conduct an ablation study on the randomness of the prompt initialization by sampling with three random seeds and report the average accuracy. For better demonstration on the sensitivity of prompt optimization performances given random initialization, we report the accuracy result with three random seeds respectively as in the following table:
>
> **Table: Accuracy (%) of randomly sampled $p_0$ with random seeds on Llama3 in 16-shot (per class) setting**
> |                                   | CoLA  | MNLI  | QNLI  | SNLI  | WNLI  |
> |-|---|-------|-------|-------|-------|
> | random $p_0$ (seed = 14) | 50.43 | 35.30 | 53.21 | 37.41 | 56.34 |
> | random $p_0$ (seed = 42) | 37.20 | 31.91 | 53.49 | 37.09 | 43.66 |
> | random $p_0$ (seed = 81) | 43.53 | 30.91 | 53.41 | 36.16 | 59.15 |
> |Zo-PoG (ours)             | 48.13 | 35.33 | 55.90 | 38.26 | 59.15 |
>
> Moreover, we conducted additional experiments on RoBERTa-Large under the same setup.
>
> **Table: Accuracy (%) of randomly sampled $p_0$ with random seeds on RoBERTa-Large in 16-shot (per class) setting**
> |                                   | CoLA       | MNLI       | QNLI       | SNLI       | WNLI       |
> |-|-|-|-|-|-|
> | random $p_0$ (seed = 14) | 52.64 |	27.62 |	54.57 |	39.06 |	47.89 |
> | random $p_0$ (seed = 42) | 46.69 |	28.97 |	53.14 |	39.65 |	52.11 |
> | random $p_0$ (seed = 81) | 55.23 |	26.25 | 	51.18 |	38.32 |	42.25    |
> | ZO-PoG (Ours)                     | 56.79 | 35.11 | 55.44 | 39.12 | 53.99 |

---

> ### Author Response · Authors · 2024-12-02
> **Responses to the follow-up questions [Part 2]**
>
> In order to further verify the sensitivity of prompt initialization, we also tried the following different sampling strategies for $p_0$: 1) we sample the same token for each position of the fixed length prompt. 2) we uniformly randomly sample from the vocabulary (the same way as that sampled with random seeds). 3) We sample the continuous tokens of fixed length from the PTM vocabulary (this is the same way as implemented in BBT). We have also tried to directly sample $p_0$ in the embedding space as initialization through 3) Gaussian distribution (with the same mean and variance value for the sampling of random matrix $A$) and 4) uniform distribution within a fixed region (mean $\pm$ standard deviation of the sampling of random matrix $A$). For these ablations, we report the accuracy with three different random seeds. 6) Orthogonal initialization: each token embedding in $p_0$ are orthogonal with each other.
>
> **Table: Accuracy (%) on RoBERTa-Large in 16-shot (per class) setting with different initialization strategy**
> |                               | CoLA  |       |       | MNLI  |       |       | QNLI  |       |       | SNLI  |       |       | WNLI  |       |       |
> |-|-|-|-|-|-|-|-|-|-|-|-|-|-|-|-|
> | **seed**                      | 14    | 42    | 81    | 14    | 42    | 81    | 14    | 42    | 81    | 14    | 42    | 81    | 14    | 42    | 81    |
> | prompt with same tokens       | 46.4  | 44.87 | 45.93 | 28.14 | 26.16 | 27.58 | 55.3  | 52.17 | 52.7  | 36.99 | 40.06 | 38.59 | 42.25 | 42.25 | 38.03 |
> | Uniform sampling (Vocabulary) | 52.64 | 46.69 | 55.23 | 27.62 | 28.97 | 26.25 | 54.57 | 53.14 | 51.18 | 39.06 | 39.65 | 38.32 | 47.89 | 52.11 | 42.25 |
> | continuous token in vocabulary | 45.54 | 49.19 | 42.57 | 26.93 | 32.59 | 33.16 | 51.38 | 54.37 | 52.00 | 38.91 | 35.15 | 34.33 | 42.25 | 56.33 | 53.52 |
> | Gaussian sampling (embedding) | 50.24 | 49.28 | 53.4  | 32.49 | 34.02 | 33.14 | 52.04 | 52.44 | 53.96 | 35.38 | 35.87 | 36.23 | 45.07 | 50.7  | 42.25 |
> | Uniform sampling (embedding)  | 48.03 | 43.53 | 54.75 | 35.24 | 32.21 | 33.06 | 52.39 | 50.52 | 51.33 | 38.84 | 34.22 | 33.58 | 49.3  | 52.11 | 54.93 |
> | Orthogonal initialization      | 42.57 | 37.87 | 53.98 | 33.03 | 32.64 | 33.42 | 53.54 | 52.02 | 52.83 | 37.38 | 34.00 | 35.01 | 54.93 | 45.07 | 54.93 |
>
> For all the above ablation studies, $p_0$ will be fixed during the optimization process and only the low dimensional soft prompt will be optimized. The above results state that performance of black-box prompt learning is sensitive to the random initialization, which motivates us to optimize the initial prompt distribution.
>
> For Weakness 4, we need to clarify that our ZO-PoG alternatively conduct discrete optimization and continuous optimization iteration by iteration. Whereas "one round + one round" optimization strategy means that we first optimize soft prompt $z$ while keep $p_0$ fixed (this is the first round), and then we optimize $p_0$ while keep soft prompt fixed (this is the second round). Both optimization strategies are under the same budget limit for fair comparison. In the final version of the manuscript, we will include the corresponding discussions of the interpretation of different optimization strategies.

---

> > ### Author Response · Authors · 2024-12-02
> > **Could you please kindly let us know if our response answers your follow-up questions?**
> >
> > Dear Reviewer hHWU,
> >
> > Thank you for your thoughtful review and follow-up questions. We have provided futher clarifications regarding our main contributions of this work and more ablations as a response to your questions. Could you please kindly let us know if our reponse resolved your futher questions? Your further feedback would be greatly appreciated.
> >
> > Best regards,
> > The Authors

---

> ### Author Response · Authors · 2024-12-03
>
> Dear Reviewer hHWU,
>
> We sincerely appreciate your willingness to engage in discussions with us and for your suggestion of our paper.
>
> We understand your concern about the extension to the black-box large language models. However, we don't think this will hurt the novelty of this work.  Our framework, ZO-PoG, uniquely integrates discrete and continuous prompt optimization in a collaborative manner. By alternating between discrete prompts optimized via policy gradients and continuous prompts refined through zeroth-order gradients, we bridge gaps in adaptability and efficiency that are not addressed in prior methods. The novelty of our method lies in the collaborative optimization of discrete and continuous prompts using policy gradient and zeroth-order techniques. This fundamental contribution remains unchanged, irrespective of the model type. Instead, extending the application of our framework to the black-box large language model setting could enhance the work’s robustness and scalability.
>
>
> Thanks again for your suggestion. Providing the results obtained from using different seeds and various sampling methods indeed enhances the motivation of our paper. Since modifications to the current manuscript are not permitted, we commit to including additional discussions and graphical presentations of these results in future revised versions.
>
> We apologize for the manner in which the results were presented in the Llama3 table in our previous response, as this presentation may have caused misunderstandings. In the previous Llama3 table, to illustrate the sensitivity of prompt performance to random seeds, we separately reported the results of runs for each seed using the method of initializing $p_0$ with random sampling, as well as the average performance over **multiple runs** using our method. **This biased comparison created the false impression that initializing with random sampling yields the highest performance.** To clarify this misunderstanding, we provide the results of our method across various runs and offer a fair comparison based on the maximum and average values, as shown in the table below. It is evident that our method outperforms random sampling initialization in both average and maximum values, thereby demonstrating its effectiveness. Regarding your mention of optimizing random seeds, we don't think this is a fair comparison to our method, as it equates to comparing the maximum performance achieved through random initialization with the average performance of our method.
>
> **Table: Accuracy (%) on Llama3 in 16-shot (per class) setting**
>
> | | CoLA | MNLI | QNLI | SNLI | WNLI |
> |---------------|-----------|-----------|-----------|-----------|-----------|
> |  random initialization (seed 14) |   50.43   |   35.30   |   53.21   |   37.41   |   56.34   |
> |  random initialization (seed 42) |   37.20   |   31.91   |   53.49   |   37.09   |   43.66   |
> |  random initialization (seed 81) |   43.53   |   30.91   |   53.41   |   36.16   |   59.15   |
> | Zo-POG (first run) |   41.71   |   35.38   |   53.14   |   39.44   |   56.34   |
> | Zo-POG (second run) |   49.28   |   35.36   |   55.67   |   35.01   |   57.75   |
> | Zo-POG (third run) |   53.40   |   35.24   |   58.91   |   40.33   |   63.38   |
> |  **random initialization (Max)** | **50.43** | **35.30** | **53.49** | **37.41** | **59.15** |
> |         **Zo-POG (MAX)**         | **53.40** | **35.38** | **58.91** | **40.33** | **63.38** |
> | **random initialization (Mean)** | **43.72** | **32.71** | **53.37** | **36.89** | **53.05** |
> |         **Zo-POG (Mean)**        | **48.13** | **35.33** | **55.90** | **38.26** | **59.15** |
>
>
> Best regards,
>
> The Authors

---

### Official Review · Reviewer_PhZG · 2024-11-03

**Soundness:** 4
**Presentation:** 3
**Contribution:** 3
**Rating:** 8
**Confidence:** 3

**Summary:**

In the paper "Collaborative Discrete-Continous Black-Box Prompting Learning For Language Model," the authors propose a new method for learning prompts, considering LLMs as a black box model. Under this assumption, the algorithms only have access to token and output (thus possible computation of a loss); therefore, it is impossible to access gradient and modify embeddings contrary to classical prefix tuning.



Similar to the CMA-ES approach, the authors propose to learn prompt embeddings in a low dimension and then project it to the input dimension of the model. The input embeddings correspond to the addition of the continuous prompt and representation of tokens of the vocabulary  $Az + p_0$($p_0$ sampled from the vocabulary). Contrary to previous approaches, authors propose to learn $z$ using the zeroth-order approximation.
In addition, the authors propose to learn the distribution of discrete prompt tokens over the vocabulary (sampling p_0). The distribution is approximated using the gamble softmax distribution, where parameters are estimated using a policy gradient with a baseline (REINFORCE).

The authors propose to analyze the convergence of the algorithm proposed in section 4. The approach is then evaluated on different GLUE tasks, and performances are compared with the manual prompt approach (designing the tokens of the prompts manually) and other black box prompting approaches. Three LLMs are compared: RoBERTa large, GPT2-XL, and Llama3.
An ablation study is set up, removing the Gumbel softmax (using a policy gradient approach from [1] instead), removing the discrete prompts optimization and the continuous prompt optimization.

For all experiments, the settings were tested at 20 and 50 prompt lengths.

The contributions are the following :
 * A new algorithm for black box prompting.
 * A new approach to select (sample) $p_0$ for discrete prompts  using  Gumbel softmax (contrary to the previous approach, choosing random token vocabulary)
* State-of-the-art results in black box prompting.
* Convergence analysis of the proposed method

**Strengths:**

* The approach and related works are well-described and motivated
* New algorithm  for black box prompting combining previous ideas with new one proposed
* Proposal of optimizing discrete prompt using gumbel softmax
* State-of-the-art result for approach in a black box setting
* Relevant Ablation study removing some part of the algorithm to judge the effect of the different component
 * Justification of the algorithm

**Weaknesses:**

* Limited dataset and configuration (20 and 50-length prompt) evaluation
* The related works section could have been extended

**Questions:**

* In the 5.1 section, the authors observed a significant difference depending on the model when compared to the manual prompt (first result line of the tables). Particularly, GPT2-XL and mostly Llama3 have, for most datasets, lower improvement using black box rather than manual prompt, particularly on the CoLA dataset. The authors state that it is due to the nature of the CoLA corpus. Does the fact that Roberta is an encoder only and the others are decoders (different pretraining tasks) have a role in this difference? The latter are thus more likely to get better performances on grammatically correct prompts.
* What is the prompt size for manual prompts? How did you select manual prompts?

---

> ### Author Response · Authors · 2024-11-22
> **Response to Reviewer PhZG**
>
> We appreciate the time and effort put into reviewing our manuscript. Please find our responses to your concerns below:
>
> **weakness 1:** "Limited dataset and configuration (20 and 50-length prompt) evaluation":
>
> Thank you for this question. We have conducted an ablation study on the effect of th prompt length. The results in the following table show that the relationship between prompt length and model performance is non-linear. Short prompts may demonstrate limited representational capacity, leading to suboptimal performance and longer prompts do not necessarily produce better result. The observed trends suggest that a moderately long prompt provides a good trade-off between capacity and learnability.
>
> ### Table: Ablation study of prompt lengths on Llama3 in 16-shot (per class) setting with the QNLI dataset
>
> | Dataset | 10   | 20   | 30   | 40   | 50   | 80   |
> |---------|-------|-------|-------|-------|-------|-------|
> | QNLI    | 53.17 | 55.90 | 56.13 | 55.83 | 55.60 | 54.73 |
>
> **weakness 2:** Thank you for your suggestions. In the revised version, we have added some more recent work.
>
> **question 1:** We thank the reviewer for the insightful observation regarding the differences in performance between models when comparing black-box prompts to manual prompts, particularly on the CoLA dataset. We also appreciate the suggestion to consider the potential influence of architectural differences between encoder-only models (e.g., RoBERTa) and decoder-only models (e.g., GPT2-XL and Llama3). In the revised manuscript, we have expanded our discussion about this point as follows: When using GPT2-XL and Llama3 as backbone models on the CoLA dataset, all prompt learning methods perform worse than manual prompts. This can be attributed to the CoLA task’s focus on grammatical acceptability, which aligns better with the pretraining objectives of encoder-only models like RoBERTa. Decoder-only models like GPT2-XL and Llama3, pretrained for generative tasks, are less sensitive to grammatical correctness unless explicitly encoded in the prompt.
> Additionally, learned prompts, such as those from BDPL, may include grammatically incorrect tokens, further impairing the performance of decoder-only models. In contrast, manual prompts, designed with grammatical correctness, provide clearer guidance for these models, explaining their superior performance on CoLA.
>
> **question 2:** The details of manual prompts template are included in Appendix B.1 with no additional learnable prompts, which are chosen by following previous baselines.
>
> We sincerely appreciate the reviewer’s insightful feedback. If there are any additional concerns or questions, please feel free to let us know.

---

> ### Author Response · Authors · 2024-11-29
>
> Dear Reviewer PhZG,
>
> Thank you again for your time and effort dedicated to reviewing our paper. As the discussion phase is nearing its end, please let us know if our responses have addressed your concerns. If you have remaining concerns or questions, please don't hestitate to let us know and we are happy to discuss with you.
>
> Best regards,
>
> The Authors

---

> > ### Comment · Reviewer_PhZG · 2024-12-02
> >
> > I acknowledge the authors' response and thank them for the additional information provided. Mainly, the authors have answered my question/remarks by providing additional experiments, addressing empirical evaluation issues, and expanding the related works part.
> >  The authors also addressed other reviewers' concerns, particularly on joint and alternate optimization, prompt initialization, and queries. That said, I agree with reviewer hHWU that the black-box models rarely provide access to token embeddings, and thus, selecting $p_0$ from vocabulary embeddings is not always possible.
> >  In the end, the authors thoughtfully answered the different questions; I believe the paper should be accepted at the ICLR conference.

---

> > > ### Author Response · Authors · 2024-12-02
> > > **Thank you for acknowledging our research**
> > >
> > > Dear Reviewer PhZG,
> > >
> > > We greatly appreciate your acknowledgement to our work and we are happy that our response has addressed your questions.

---

### Official Review · Reviewer_yHre · 2024-11-05

**Soundness:** 3
**Presentation:** 3
**Contribution:** 2
**Rating:** 5
**Confidence:** 3

**Summary:**

This paper proposes a new approach to black-box prompt learning by jointly optimizing both discrete text prompts and continuous embeddings, with a convergence analysis. Experiments on five commonly used datasets demonstrate its superiority.

**Strengths:**

- The paper is clearly written and easy to follow
- The proposed method is technically sound
- The experimental results show the proposed method has a significant performance improvement.

**Weaknesses:**

- My main concern lies in that why the authors did not employ the true black-box models, such as GPT-4 and Claude-3.5, as their backbone models. Since the proposed method aims at black-box prompt learning, it would be more convincing to show how it works with these leading black-box API.  Is it still necessary to optimize the prompts for such powerful models in terms of cost versus benefit? It would be much better to include these results.
- Why choosing a random matrix $\textbf{A}$? Is it good enough? If not, why not optimizing it or how to choose a better one?
- How does the performance change as the prompt length increases? Better show more results on different lengths.
- Does this method works only on natural language tasks? How does it behave on math or code tasks?

**Questions:**

See the weaknesses.

---

> ### Author Response · Authors · 2024-11-22
> **Response to Reviewer yHre**
>
> We appreciate the time and effort put into reviewing our manuscript. Please find our responses to your concerns below:
>
> **weakness 1:** We appreciate your insightful suggestion regarding the evaluation of our proposed method on truly black-box models such as GPT-4. In the revised manuscript, we have adapted our method to the real black-box large language models. To tackle the problem having no access to the embedding space of black-box LLMs, we introduce to use an open-sourced embedding model to construct the mapping between the whole vocabulary and their embedding. To be specific, let $V_{s}$ denote the full vocabulary of the surrogate embedding model. Before the training of the algorithm, we will first need to compute the embedding ${ e(v) | v \in V_s }$ for all tokens in $V_{s}$. Then we could construct a lookup table $\{(v, e(v)) \mid v \in V_s\}$ for efficient access during training. During training, we define a projection function $e^{-1}(\cdot): R^D \to V_s^n$ that map a learned embedding $p \in R^D$ back to the nearest or the most similar token in $V_s$. In this work, we choose the cosine similarity as the projection function, i.e.,
> $$
>     e^{-1} (p) =  argmax_{v \in V_s^n} \frac{\< p, e(v)\>}{|| p ||  || e(v)||},
> $$
> which is a token-wise projection operation. By projecting the soft prompt back to the token space, we reformulated our approach to operate purely within the constraints of API-based interactions. To demonstrate the practicality and effectiveness of the updated method, we conducted additional experiments using GPT-4. The results, included in Appendix D.2 of the revised manuscript, show that our method achieves competitive performance on the mathematical reasoning task GSM8K while requiring no access to embeddings or model internals. This extension strengthens the significance of our work by aligning it with the limitations of modern commercial LLM APIs.
>
> ### Table: Averaged scores of baselines and Zo-PoG on the GSM8K dataset using GPT-4
>
> | Dataset | MP CoT | BBT   | BDPL  | TRIPLE | InstructZero | Zo-PoG (ours) |
> |---------|--------|-------|-------|--------|--------------|---------------|
> | GSM8K   | 86.48  | 78.22 | 85.77 | 70.26  | 89.16        | 89.94         |
>
> **weakness 2:** Thank you for pointing this problem out.
>
> ``why choosing a random matrix?": In our framework, we choose a random projection matrix  $A$  to efficiently explore the intrinsic subspace of the embedding space. This choice is motivated by its simplicity and computational efficiency, which make it well-suited for the black-box optimization setting.
>
> "Is it good enough?": Random projections have been shown to work well in many scenarios, as observed in our experiments.
>
> "why not optimizing it?": The original embedding space of pre-trained language models is often extremely high-dimensional, which makes direct black-box optimization computationally prohibitive. Optimizing  $A$, which is of dimension $R^{D\times d}$, introduces additional computational overhead and may complicate the optimization process.
>
> "How to choose a better one?": Actually, the baseline method SSPT we compared in the experiment explored choosing a better one by
> employing subspace learning to identify the optimal ultra-low-dimensional subspace. Considering the optimization complexity for $A$, we instead utilize a random, which already yields good performance in practice.
>
> **weakness 3:**
> We appreciate the reviewer’s insightful suggestion regarding the impact of prompt length on performance. In the revised version, we have conducted an ablation study exploring the effect of prompt length on model performance. The results indicate that a moderate length prompt is optimal and longer prompts do not necessarily produce better results. The results, along with more detailed analysis, are included in the revised manuscript in Appendix section C.
>
> ### Table: Ablation study of prompt lengths on Llama3 in 16-shot (per class) setting with the QNLI dataset
>
> | Dataset | 10   | 20   | 30   | 40   | 50   | 80   |
> |---------|-------|-------|-------|-------|-------|-------|
> | QNLI    | 53.17 | 55.90 | 56.13 | 55.83 | 55.60 | 54.73 |
>
> **weakness 4:**
> Thanks for this question. In the revised version, we have conducted experiments on the GSM8K dataset to show our algorithm performance on the math problems. In this experiment, we used GPT-4 to adapt our algorithm to a true black-box model.
>
> We sincerely appreciate the reviewer’s insightful feedback. If there are any additional concerns or questions, please feel free to let us know.

---

> > ### Author Response · Authors · 2024-11-25
> >
> > Dear reviewer, thank you again for your time and effort put into reviewering our manuscript. Please let us know if our responses have addressed your concerns. If you have remaining concerns, please don't hestitate to let us know and we are happy to address them. Thank you!

---

> ### Author Response · Authors · 2024-11-27
>
> Dear Reviewer yHre,
>
> Thank you again for your time and effort dedicated to reviewing our paper. Please let us know if our responses have addressed your concerns. If you have remaining concerns, please don't hestitate to let us know and we are happy to address them.
>
> With best wishes,
>
> Authors

---

> ### Author Response · Authors · 2024-11-29
>
> Dear Reviewer yHre,
>
> Thank you again for your time and effort dedicated to reviewing our paper. As the discussion phase is nearing its end, please let us know if our responses have addressed your concerns. If you have remaining concerns or questions, please don't hestitate to let us know and we are happy to discuss with you.
>
> Best regards,
>
> The Authors

---

### Official Review · Reviewer_sg5f · 2024-11-07

**Soundness:** 3
**Presentation:** 3
**Contribution:** 3
**Rating:** 6
**Confidence:** 3

**Summary:**

This paper introduces a framework for prompt optimization in LLMs that combines discrete and continuous prompt tuning. This framework alternates between optimizing discrete prompts through policy gradient methods and continuous prompts via zeroth-order gradient optimization. The authors establish the sub-linear convergence of ZO-PoG under assumptions of Smoothness, Bounded Variance, Bounded Loss, and Lower-Bounded Parameters. This framework was evaluated on five datasets (CoLA, MNLI, QNLI, SNLI, WNLI) from the GLUE benchmark using RoBERTa-large, GPT2-XL, and Llama3 as backbone models. The code has been provided.

**Strengths:**

1. ZO-PoG combines discrete and continuous prompt optimization. Discrete prompts are refined through policy gradient in the parameter space, and continuous prompts are adjusted using zeroth-order gradients. This dual optimization approach enhances the adaptability and efficiency of prompt learning.
2. The authors provide a theoretical analysis showing ZO-PoG’s sub-linear convergence, validating its efficiency. Sub-linear convergence means that ZO-PoG requires fewer iterations to achieve satisfactory performance, making it a computationally efficient solution for black-box prompt learning.
3. The framework is tested on five datasets and demonstrates improvements over other black-box prompt learning methods. Results from the ablation study confirm that each component positively contributes to ZO-PoG’s overall performance, further validating its design choices.

**Weaknesses:**

1. Assumption 3 requires the loss function to be bounded. Does this assumption hold for the loss function in LLM fine-tuning? If not, could this assumption be relaxed in the theoretical analysis presented in this paper?
2. This paper evaluates ZO-PoG on five datasets from the GLUE benchmark, while methods like BBT and SSPT report results on additional datasets. It would be valuable if the authors could provide performance comparisons of ZO-PoG on a broader range of datasets to offer a more comprehensive evaluation.
3. I recommend that the authors discuss the query complexity of zeroth-order optimization following the theorem, as this would provide a clearer understanding of the computational efficiency of the proposed approach.

**Questions:**

Given that the loss function of LLMs is unbounded, I'm wondering if Assumption 3 can be relaxed in theoretical analysis to better reflect practical scenarios?

---

> ### Author Response · Authors · 2024-11-22
> **Response to Reviewer sg5f**
>
> We appreciate the time and effort put into reviewing our manuscript. Please find our responses to your concerns below:
>
> **weakness 1:** Thank you for pointing out this important consideration regarding the boundedness of the loss function. While the cross-entropy loss used in our work is theoretically unbounded, in practice, it behaves as effectively bounded during large language model (LLM) fine-tuning. This is due to numerical stability in softmax calculations, and the finite precision of modern computing hardware, which prevent predicted probabilities from reaching exactly zero. Furthermore, empirical observations across LLM fine-tuning tasks consistently show that the loss remains within a stable and reasonable range, which aligns with the boundedness assumption.
>
> **weakness 2:** Thanks for your suggestion. In the revised version, we have included more experimental results to verify the effectiveness of ZO-PoG. Specifically, we have adapted our method to the real black-box large language models. To tackle the problem having no access to the embedding space of black-box LLMs, we introduce to use an open-sourced embedding model to construct the mapping between the whole vocabulary and their embedding. To be specific, let $V_{s}$ denote the full vocabulary of the surrogate embedding model. Before the training of the algorithm, we will first need to compute the embedding ${ e(v) | v \in V_s }$ for all tokens in $V_{s}$. Then we could construct a lookup table $\{(v, e(v)) \mid v \in V_s\}$ for efficient access during training. During training, we define a projection function $e^{-1}(\cdot): R^D \to V_s^n$ that map a learned embedding $p \in R^D$ back to the nearest or the most similar token in $V_s$. In this work, we choose the cosine similarity as the projection function, i.e.,
> $$
>     e^{-1} (p) =  argmax_{v \in V_s^n} \frac{\< p, e(v)\>}{|| p ||  || e(v)||},
> $$
> which is a token-wise projection operation. By projecting the soft prompt back to the token space, we reformulated our approach to operate purely within the constraints of API-based interactions. To demonstrate the practicality and effectiveness of the updated method, we conducted additional experiments using GPT-4. The results, included in Appendix D.2 of the revised manuscript, show that our method achieves competitive performance on the mathematical reasoning task GSM8K while requiring no access to embeddings or model internals. This extension strengthens the significance of our work by aligning it with the limitations of modern commercial LLM APIs.
>
> ### Table: Averaged scores of baselines and Zo-PoG on the GSM8K dataset using GPT-4
>
> | Dataset | MP CoT | BBT   | BDPL  | TRIPLE | InstructZero | Zo-PoG (ours) |
> |---------|--------|-------|-------|--------|--------------|---------------|
> | GSM8K   | 86.48  | 78.22 | 85.77 | 70.26  | 89.16        | 89.94         |
>
> **weakness 3:** Thank you for your suggestion, in the revised version, we have added the corresponding discussions about the query complexity. To be specific, the theorem establishes that ZO-PoG achieves an $\epsilon$-stationary point in expectation with total query complexity ( times of forward passes of the PTMs) $T\cdot I = \mathcal{O} \left( \frac{\sqrt{n\kappa}}{\epsilon^3} \right)$.
>
> **questions:** As discussed above in response to weakness 1, we think the bounded loss assumption is a general assumption and is reasonable in practical scenarios.
>
> We sincerely appreciate the reviewer’s insightful feedback. If there are any additional concerns or questions, please feel free to let us know.

---

> ### Author Response · Authors · 2024-11-29
>
> Dear Reviewer sg5f
>
> Thank you again for your time and effort dedicated to reviewing our paper. As the discussion phase is nearing its end, please let us know if our responses have addressed your concerns. If you have remaining concerns or questions, please don't hestitate to let us know and we are happy to discuss with you.
>
> Best regards,
>
> The Authors

---

> > ### Comment · Reviewer_sg5f · 2024-12-03
> > **Thanks for the rebuttal**
> >
> > Thank you for your response and for including the query complexity results. I will keep my score.

---

### Author Response · Authors · 2024-11-27
**Summarization of Paper Revisions**

Dear Reviewers,

We appreciate your thorough evaluations and constructive feedback on our submission. We have carefully addressed all your comments and made the following revisions to improve the paper:

1. **Extended related work:** Reviewer PhZG suggested that the related work section could be extended. In the revised version, we have revised this part and included more recent work on black-box prompt learning.

2. **Discussions on query complexity:** As suggested by Reviewer sg5f, we added the discussion about the query complexity (here we refer to the times of forward passes of the pre-trained language models) of the proposed ZO-PoG method for converging to an $\epsilon$-approximate stationary point.

3. **Discussions on experimental results:** We sincerely appreciate Reviewer PhZG's suggestion to consider the potential influence of architectural differences between encoder-only models (e.g., RoBERTa) and decoder-only models (e.g., GPT2-XL and Llama3) as the factor of performance gap on CoLA dataset when compared with the manual prompt. We have added the corresponding discussions in the revised section 5.2.

4. **Additional ablation studies** According to reviewers' suggestions and concerns, we have conducted additional ablation studies in Appendix Section C, including the effect of optimizing initial prompt (Reviewer hHWU), the effect of prompt length (Reviewer yHre, Reviewe PhZG) and the effect of optimization strategy (Reviwer hHWU).

5. **Additional experiments:** Reviewers yHre and hHWU had the concerns of our method on real black-box language models, which only accept prompt tokens as input. In response, we have extended our method to this more challenging setting in Appendix Section D. The core idea is to incorporate an open-sourced embedding model to project continuous embedding back to the discrete token space. To verify the effectiveness of ZO-PoG on more complex tasks with black-box language models. We have conducted experiments on the mathematical reasoning task GSM8K.

We appreciate the time and effort you have invested in reviewing our work. Your comments have significantly strengthened our submission, and we hope that the revisions address your concerns comprehensively.

Thank you again for your thoughtful feedback and consideration.

Best regards,

The Authors

---

### Meta-Review · Area_Chair_V9qY · 2024-12-20

**Metareview:**

This paper introduces a new framework for prompt optimization in black-box LLMs that combines discrete and continuous prompt tuning using policy gradient and zeroth-order gradient optimization methods. The authors prove sub-linear convergence under certain conditions and demonstrate the effectiveness of their approach on five GLUE benchmark datasets.

The reviewers generally appreciated that the paper is well-written and technically sound. The proposed ZO-PoG method enhances prompt learning adaptability and efficiency. The theoretical analysis provided further supports the method’s effectiveness. Although several questions and concerns were raised during the initial round of reviews, including: 1) whether the assumption of a bounded loss function holds for LLM fine-tuning, 2) the absence of real black-box LLMs in the study, 3) the limited number of evaluation datasets and the recommendation to compare with other methods, and 4) concerns about the choice of a random matrix, the need for alternating optimization, and the impact of prompt length, the authors have addressed most of these points in the rebuttal. The remaining major concern comes from reviewer hHWU, who questions the novelty of the approach, noting that the idea of steering a white-box model’s embedding and constructing an embedding lookup table is similar to existing works such as InstructZero and ZOPO. However, the authors argue that the primary novelty of this work lies in its solution to the performance gap induced by the random initialization of $p_0$, as well as its collaborative optimization framework using policy gradient and zeroth-order techniques, which is also recognized as innovative by hHWU. After careful reading of the paper and discussion, I agree with the authors on this point. Given that all other major concerns have been addressed, I recommend acceptance. However, I strongly suggest that the authors include the additional discussions and experiments in the final version.

**Additional Comments On Reviewer Discussion:**

The reviewers initially expressed concerns about 1) whether the assumption of a bounded loss function holds for LLM fine-tuning, 2) the absence of real black-box LLMs in the study, 3) the limited number of evaluation datasets and the recommendation to compare with other methods, and 4) the choice of a random matrix, the need for alternating optimization, and the impact of prompt length. In response, the authors provided additional clarifications and experiments, including tests with different prompt lengths, results using real black-box LLMs (e.g., GPT-4), new results on the Math dataset GSM8K, and accuracy comparisons with randomly sampled $p_0$ across different seeds. These additional details addressed most concerns, though reviewer hHWU raised a novelty issue, noting that the idea of steering a white-box model’s embedding is similar to InstructZero, and constructing an embedding lookup table resembles work in ZOPO. Acknowledging these points, the authors argue that the primary focus and novelty of the paper lies in addressing the performance gap induced by the random initialization of $p_0$. The newly proposed optimization framework, recognized by reviewer hHWU as novel, strengthens the paper's contribution. I find that the newly added analysis and experiments, particularly the comparison with random initialization, provide a stronger motivation for the work. Thus, I am recommending acceptance, assuming that all discussions and experiments are incorporated into the final version.

---

### Decision · Program_Chairs · 2025-01-22

Accept (Poster)